# INDIVIDUALISED DOSE-RESPONSE ESTIMATION USING GENERATIVE ADVERSARIAL NETS

## ABSTRACT

The problem of estimating treatment responses from observational data is by now a well-studied one. Less well studied, though, is the problem of treatment response estimation when the treatments are accompanied by a continuous dosage parameter. In this paper, we tackle this lesser studied problem by building on a modification of the generative adversarial networks (GANs) framework that has already demonstrated effectiveness in the former problem. Our model, DRGAN, is flexible, capable of handling multiple treatments each accompanied by a dosage parameter. The key idea is to use a significantly modified GAN model to generate *entire* dose-response curves for each sample in the training data which will then allow us to use standard supervised methods to learn an inference model capable of estimating these curves for a new sample. Our model consists of 3 blocks: (1) a generator, (2) a discriminator, (3) an inference block. In order to address the challenge presented by the introduction of dosages, we propose novel architectures for both our generator and discriminator. We model the generator as a multi-task deep neural network. In order to address the increased complexity of the treatment space (because of the addition of dosages), we develop a hierarchical discriminator consisting of several networks: (a) a treatment discriminator, (b) a dosage discriminator for each treatment. In the experiments section, we introduce a new semi-synthetic data simulation for use in the dose-response setting and demonstrate improvements over the existing benchmark models.

## 1 INTRODUCTION

Most of the methods developed in the causal inference literature focus on learning the effects of binary or categorical treatments (Bertsimas et al., 2017; Alaa et al., 2017; Alaa & van der Schaar, 2017; Athey & Imbens, 2016; Wager & Athey, 2018; Yoon et al., 2018). These treatments, though, are often administered at a certain dosage which can take on continuous values (such as vasopressors (Döpp-Zemel & Groeneveld, 2013)). In medicine, using a high dosage of a drug can lead to toxic effects while using a low dosage can result in no effect on the patient outcome (Wang et al., 2017). Moreover, the dosage levels used when choosing between multiple treatments for a patient are crucial for the decision (Rothwell et al., 2018).

While admissible dosage intervals for drugs are often determined from clinical trials (Cook et al., 2015), these trials often have a small number of patients and use simplistic mathematical models to assign dosage levels to patients that do not take into account patient heterogeneity (Ursino et al., 2017). After drugs are approved through clinical trials, observational data collected about different treatment dosages prescribed to a diverse set of patients offers us the opportunity to learn individualized responses. As the relationships between treatment dosage efficacy, toxicity and patient features become more complex, estimating dose-response from observational data becomes particularly important in order to identify optimal dosages for each patient. Fortunately, there is a wealth of observational data available in the medical domain from electronic health records (Henry et al., 2016).

Learning from observational data already presents significant challenges in the binary treatment setting. As explained by Spirtes (2009), in an observational treatment-effect dataset, only the factual outcome is present (i.e. the outcome for the treatment that was actually given) - the counterfactual outcomes are not observed. This problem is exacerbated in the dose-response setting in which the number of counterfactuals is no longer even finite. Moreover, the treatment assignment is non-random

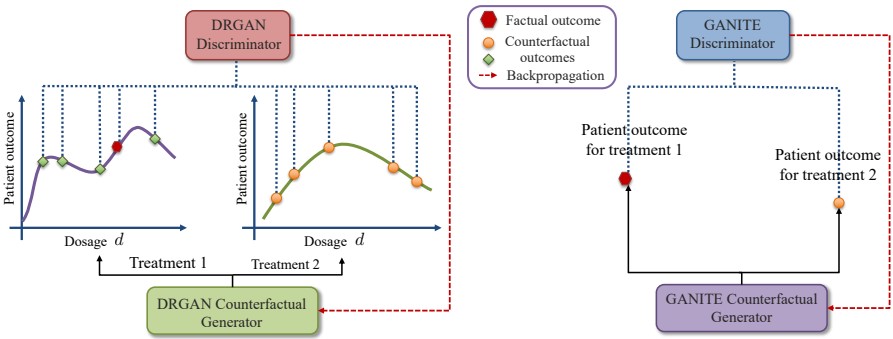

Figure 1: Comparison of DRGAN and GANITE highlighting the key difference between the two different problems they address (dose-response estimation vs. standard treatment-response estimation).

and instead is assigned according to the features associated with each sample. Due to the continuous nature of the dosage parameter, adjusting for the bias in the dosage assignments is significantly more complex than for binary (or even multiple) treatments. Thus, standard methods for adjusting for treatment selection bias cannot be easily extended to handle bias in the dosage parameter.

In this paper we address the problem of dose-response estimation from observational data by building on the framework introduced in GANITE (Yoon et al., 2018). The key idea is to modify the GAN framework (Goodfellow et al., 2014) to generate the unobserved counterfactual outcomes from a standard treatment-effect dataset. Already, GANITE presents a significant modification to the original GAN framework - rather than the discriminator discriminating between entirely real or entirely fake samples, the discriminator is attempting to pick out the real component from a vector containing the real (factual) outcome from the dataset and the fake (counterfactual) outcomes generated by the generator. We also inherit this key difference from a standard GAN, but in addition we must make further modifications to the original GANITE framework in order to address the dosage problem.

A naive attempt to extend Yoon et al. (2018) to the dosage setting might involve trying to define a discriminator that takes as input an entire dose-response curve for each treatment from the generator (with the outcome for the observed treatment-dosage pair replacing the generated one) and that tries to determine the factual treatment-dosage pair. This fails for two reasons: (1) we do not wish to assume prior knowledge of the functional form of the dose-response curves and so will have access to the generated dose-response curves only by evaluating them at given points (and so "entire" dose-response curves cannot be passed to the discriminator); (2) substituting the generator output for the factual treatment-dosage pair with the factual outcome will almost always create a discontinuity in the response curve and thus the factual treatment-dosage pair would be very easy to identify.

We overcome these two hurdles by defining a discriminator that, rather than acting on the entire dose-response curves, acts on a finite set of points from each curve, as shown in Fig. 1. From *among the chosen points*, the discriminator will then attempt to pick out the factual one. To ensure that the *entire* dose-response curve is well-estimated, we sample the set of points randomly *each* time an input would be passed to the discriminator. If we were to fix a set of points in advance to compare for *all* treatments and samples then only the outcomes associated with these dosage levels would be well estimated. As our discriminator will be taking as input a *set* of random dosage-outcome pairs, we need to condition its behaviour to be like that of a function on a set. In particular, we draw on ideas from Zaheer et al. (2017) to ensure that the discriminator acts as a function on sets and its output does not depend on the *order* in which the elements of the set are given as input.

In addition, we model the generator as a multi-task deep network capable of taking dosages as an input; this gives us the flexibility to learn heterogeneous dose-response curves for the different treatments. We also develop a *hierarchical* discriminator which breaks down the job of the discriminator into determining the factual treatment and determining the factual dosage using separate networks. We show in the experiments section that this approach significantly improves performance and is more stable than using a single network discriminator.

Our contributions in this paper are 3-fold: (1) we propose DRGAN, a significantly modified GAN framework, capable of dose-response estimation, (2) we propose novel architectures for each of our networks, (3) we propose a new semi-synthetic data simulation for use in the dose-response setting. We show, using semi-synthetic experiments, that our model outperforms existing benchmarks.

## 2 RELATED WORK

Methods for estimating the outcomes of treatments with an exposure dosage parameter that only employ observational data make use of the generalized propensity score (GPS) (Imbens, 2000; Imai & Van Dyk, 2004; Hirano & Imbens, 2004) or build on top of balancing methods for multiple treatments. Schwab et al. (2019) developed a neural network based method to estimate counterfactuals for multiple treatments and continuous dosages. The proposed Dose Response networks (DRNets) in Schwab et al. (2019) consist of a three level architecture with shared layers for all treatments, multi-task layers for each treatment and additional multi-task layers for dosage sub-intervals. More specifically, for each treatment $w$, the dosage interval $[a_w, b_w]$ is subdivided into $E$ *equally* sized sub-intervals and a multi-task head is added for each sub-interval. Their model architecture extends the one in Shalit et al. (2017) by adding the multi-task heads for the dosage strata. However, the main advantage of using multi-task heads for dosage intervals would be the added flexibility in the model to learn potentially very different functions over different regions of the dosage interval. DRNets does not determine the dosage intervals dynamically and thus much of this flexbility is lost. We demonstrate in our experiments that DRGAN outperforms both GPS and DRNets.

For a discussion of works that address treatment-response estimation without a dosage parameter, see Appendix A. Note that for such methods we cannot treat the dosage as an additional input due to the bias associated with its assignment.

## 3 PROBLEM FORMULATION

We consider receiving observations of the form $(\mathbf{x}^i, t_f^i, y_f^i)$ for $i = 1, ..., N$, where, for each $i$, these are independent realizations of the random variables $(\mathbf{X}, T_f, Y_f)$. We refer to $\mathbf{X}$ as the feature vector lying in some feature space $\mathcal{X}$, containing pre-treatment covariates (such as age, weight and lab test results). The treatment random variable, $T_f$, is in fact a pair of values $T_f = (W_f, D_f)$ where $W_f \in \mathcal{W}$ corresponds to the *type* of treatment being administered (e.g. chemotherapy or radiotherapy) which lies in the discrete space of $k$ treatments, $\mathcal{W} = \{w_1, ..., w_k\}$, and $D_f$ corresponds to the *dosage* of the treatment (e.g. number of cycles, amount of chemotherapy, intensity of radiotherapy), which, for a given treatment $w$ lies in the corresponding treatment's dosage space, $\mathcal{D}_w$, which in the most general case is some continuous space (e.g. the interval $[0, 1]$). We define the set of all treatment-dosage pairs to be $\mathcal{T} = \{(w, d) : w \in \mathcal{W}, d \in \mathcal{D}_w\}$.

Following Rubin's potential outcome framework (Rubin, 1984), we assume that for all treatment-dosage pairs, $(w, d)$, there is a potential outcome $Y(w, d) \in \mathcal{Y}$ (e.g. 1-year survival probability). The *observed* outcome is then defined to be $Y_f = Y(W_f, D_f)$. We will refer to the unobserved (potential) outcomes as *counterfactuals*.

The goal of dose-response estimation is to derive *unbiased* estimates of the potential outcomes for a given set of input covariates:

$$\mu(t, \mathbf{x}) = \mathbb{E}[Y(t)|\mathbf{X} = \mathbf{x}] \tag{1}$$

for each $t \in \mathcal{T}$, $\mathbf{x} \in \mathcal{X}$. We refer to $\mu(\cdot)$ as the individualised dose-response function. In general, this quantity is not the same as $\mathbb{E}[Y|\mathbf{X} = \mathbf{x}, T_f = t]$ (which can be easily estimated from observational data) in the presence of *selection bias* which often presents itself in observational datasets. In order for these two quantities to be equal, we must make the following common assumption.

**Assumption 1.** *(Unconfoundedness) The treatment assignment, $T_f$, and potential outcomes, $Y(w, d)$, are conditionally independent given the covariates $\mathbf{X}$, i.e.*

$$\{Y(w, d)|w \in \mathcal{W}, d \in \mathcal{D}_w\} \perp\!\!\!\perp T_f|\mathbf{X}. \tag{2}$$

*This assumption is also commonly referred to as* no hidden confounding.

In addition, in order to make $\mu(\cdot)$ identifiable we must also assume that *any* treatment-dosage pair *could* be assigned to any given sample.

**Assumption 2.** *(Overlap) For each $\mathbf{x} \in \mathcal{X}$ such that $p(\mathbf{x}) > 0$, we have $1 > p(t|\mathbf{x}) > 0$ for each $t \in \mathcal{T}$.*

## 4 DOSE-RESPONSE GAN

We propose estimating $\mu$ by first training a generator to generate dose-response curves for each sample *within* the training dataset. The learned generator can then be used to train an inference network using standard supervised methods. We build on the idea presented in Yoon et al. (2018), using a modified GAN framework to generate potential outcomes conditional on the observed features, treatment and factual outcome. Several changes must be made to both the generator and discriminator architectures and learning paradigms in order to produce a model capable of handling the dose-response setting.

### 4.1 COUNTERFACTUAL GENERATOR

Our generator, $\mathbf{G} : \mathcal{X} \times \mathcal{T} \times \mathcal{Y} \times \mathcal{Z} \to \mathcal{Y}^{\mathcal{T}}$ takes features, $\mathbf{x} \in \mathcal{X}$, factual outcome, $y_f \in \mathcal{Y}$, received treatment and dosage, $t_f = (w_f, d_f) \in \mathcal{T}$, and some noise, $\mathbf{z} \in \mathcal{Z}$ (typically multivariate uniform or Gaussian), as inputs. The output will be a dose-response curve for each treatment (as shown in Fig. 1), so that the output is a function from $\mathcal{T}$ to $\mathcal{Y}$, i.e. $\mathbf{G}(\mathbf{x}, t_f, y_f, \mathbf{z})(\cdot) : \mathcal{T} \to \mathcal{Y}$. We can then write

$$\hat{y}_{cf}(t) = \mathbf{G}(\mathbf{x}, t_f, y_f, \mathbf{z})(t) \tag{3}$$

to denote our generated counterfactual outcome for the treatment-dosage pair $t$. We will write $\hat{Y}_{cf}(t) = \mathbf{G}(\mathbf{X}, T_f, Y_f, \mathbf{Z})(t)$ (i.e. the random variable induced by $\mathbf{G}$).

While the job of the counterfactual generator is to generate outcomes for the treatment-dosage pairs which were *not* observed, Yoon et al. (2018) demonstrated that the performance of the counterfactual generator is improved by adding a supervised loss term that regularises its output for the factual treatment (in our case treatment-dosage pair). We define the supervised loss, $\mathcal{L}_S$, to be

$$\mathcal{L}_S(\mathbf{G}) = \mathbb{E}\left[(Y_f - \mathbf{G}(\mathbf{X}, T_f, Y_f, \mathbf{Z})(T_f))^2\right], \tag{4}$$

where the expectation is taken over $\mathbf{X}, T_f, Y_f$ and $\mathbf{Z}$.

### 4.2 COUNTERFACTUAL DISCRIMINATOR

As noted in Section 1, our discriminator will act on a random set of points from each of the generated dose-response curves. Similar to Yoon et al. (2018), we define a discriminator, $\mathbf{D}$, that will attempt to pick out the factual treatment-dosage pair from among the (random set of) generated ones.

Formally, let $n_w \in \mathbb{Z}^+$ be the number of dosage levels we will compare for treatment $w \in \mathcal{W}^1$. For each $w \in \mathcal{W}$, let $\tilde{\mathcal{D}}_w = \{D_1^w, ..., D_{n_w}^w\}$ be a random subset[2] of $\mathcal{D}_w$ of size $n_w$, where for the factual treatment, $W_f$, $\tilde{\mathcal{D}}_{W_f}$ contains $n_{W_f} - 1$ random elements along with $D_f$. We define $\tilde{\mathbf{Y}}_w = (D_i^w, \tilde{Y}_i^w)_{i=1}^{n_w} \in (\mathcal{D}_w \times \mathcal{Y})^{n_w}$ to be the vector of dosage-outcome pairs for treatment $w$ where

$$\tilde{Y}_i^w = \begin{cases} Y_f \text{ if } W_f = w \text{ and } D_f = D_i^w \\ \hat{Y}_{cf}(w, D_i^w) \text{ else} \end{cases} \tag{5}$$

and will write $\tilde{\mathbf{Y}} = (\tilde{\mathbf{Y}}_w)_{w \in \mathcal{W}}$. We will write $d_j^w, \tilde{\mathbf{y}}_w$ and $\tilde{\mathbf{y}}$ to denote realisations of $D_j^w, \tilde{\mathbf{Y}}_w$ and $\tilde{\mathbf{y}}$.

Our discriminator, $\mathbf{D} : \mathcal{X} \times \prod_{w \in \mathcal{W}} (\mathcal{D}_w \times \mathcal{Y})^{n_w} \to [0, 1]^{\sum n_w}$, will take the features $\mathbf{x} \in \mathcal{X}$ together with the (random) set of generated outcomes $\tilde{\mathbf{y}} \in \mathcal{Y}^{\sum n_w}$, and output a probability for each treatment-dosage pair indicating the discriminator's belief that that pair is the factual one.

As in the standard GAN framework, we define a minimax game by defining the value function to be

$$\mathcal{L}(\mathbf{D}, \mathbf{G}) = \mathbb{E}\left[ \sum_{w \in \mathcal{W}} \sum_{d \in \tilde{\mathcal{D}}_w} \mathbb{I}_{\{T_f = (w,d)\}} \log \mathbf{D}^{w,d}(\mathbf{X}, \tilde{\mathbf{Y}}) + \mathbb{I}_{\{T_f \neq (w,d)\}} \log(1 - \mathbf{D}^{w,d}(\mathbf{X}, \tilde{\mathbf{Y}})) \right], \tag{6}$$

where the expectation is taken over $\mathbf{X}, T_f, \tilde{\mathbf{Y}}$ and $\{\tilde{\mathcal{D}}_w : w \in \mathcal{W}\}$, $\mathbf{D}^{w,d}$ corresponds to the discriminator output for treatment-dosage pair $(w, d)$.

---

[1]In practice we set all $n_w$ to be the same. The default setting is 5 in the experiments.

[2]In practice, when $\mathcal{D}_w = [0, 1]$, each $D_j^w$ is sampled independently and uniformly from $[0, 1]$. Note that for each training iteration, $\tilde{\mathcal{D}}_w$ is resampled (see Section 1).

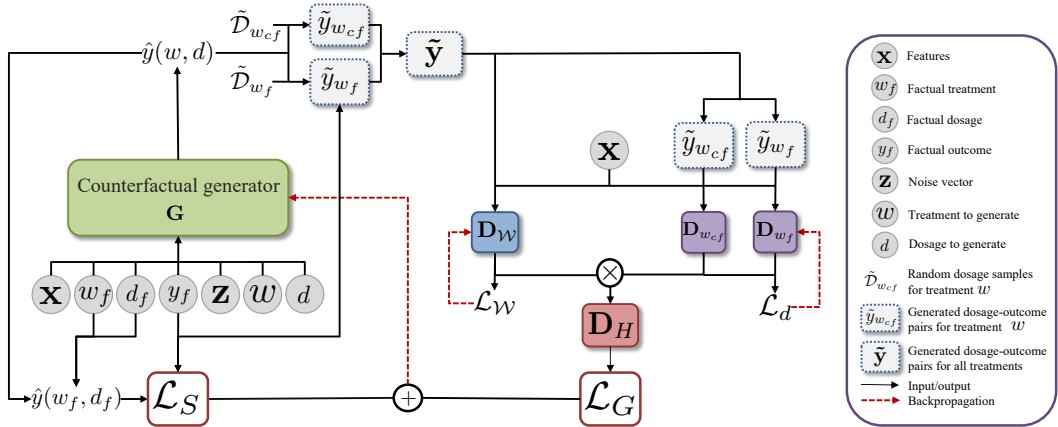

Figure 2: Overview of our model for the setting with two treatments ($w_f$ corresponds to the factual treatment and $w_{cf}$ to the counterfactual treatment). The generator is used to generate an output for each dosage level in each $\tilde{\mathcal{D}}_w$, these outcomes together with the factual outcome, $y_f$, are used to create the set of dosage-outcome pairs, $\tilde{\mathbf{y}}$, which is passed directly to the treatment discriminator. Each dosage discriminator receives only the part of $\tilde{\mathbf{y}}$ corresponding to that treatment, i.e. $\tilde{\mathbf{y}}_w$. These discriminators are combined (Eq. 11) to define $\mathbf{D}_H$ which is used to give feedback to the generator.

The minimax game is then given by

$$\min_{\mathbf{G}} \max_{\mathbf{D}} \mathcal{L}(\mathbf{D}, \mathbf{G}) + \lambda \mathcal{L}_S(\mathbf{G}), \tag{7}$$

where $\lambda$ is used to control the trade-off between $\mathcal{L}$ and $\mathcal{L}_S$ (we set $\lambda = 1$ in the experiments).

The task of the discriminator (i.e. picking out the factual dosage from $\sum_{j=1}^{k} n_{w_j}$ treatment-dosage pairs) becomes increasingly difficult as we increase $n_w$ or $k$ because the dimension of the discriminator output space, $\sum n_w$, increases. Although we control $n_w$, if we set it too low, then the set $\tilde{\mathbf{y}}_w$ may not well-represent the dose-response curve, particularly if the dose-response curve is complex. In practice we found that even for moderate settings of $n_w$ and only 2 treatments, modelling the discriminator as a single function resulted in poor performance. In order to overcome this problem, we introduce a novel hierarchical discriminator which involves a treatment discriminator with output dimension $k$ and several dosage discriminators, one for each treatment, with output dimensions $n_w$.

First observe that the probability $\mathbb{P}((W_f, D_f) = (w, d)|\mathbf{X}, \tilde{\mathcal{D}}_w, \tilde{\mathbf{Y}})$ can be written as

$$\mathbb{P}(W_f = w|\mathbf{X}, \tilde{\mathcal{D}}_w, \tilde{\mathbf{Y}}) \times \mathbb{P}(D_f = d|W_f = w, \mathbf{X}, \tilde{\mathcal{D}}_w, \tilde{\mathbf{Y}}). \tag{8}$$

We can therefore break down the discriminator into a hierarchical model by learning one discriminator, $\mathbf{D}_{\mathcal{W}}$, that outputs $\mathbb{P}(W_f = w|\mathbf{X}, \tilde{\mathcal{D}}_w, \tilde{\mathbf{Y}})$ which we will refer to as the *treatment* discriminator, and then a discriminator, $\mathbf{D}_w$, for each treatment, $w \in \mathcal{W}$, that outputs $\mathbb{P}(D_f = d|W_f = w, \mathbf{X}, \tilde{\mathcal{D}}_w, \tilde{\mathbf{Y}})$ which we will refer to as the *dosage* discriminator for treatment $w$.

The treatment discriminator, $\mathbf{D}_{\mathcal{W}} : \mathbf{X} \times \prod_{w \in \mathcal{W}} (\mathcal{D}_w \times \mathcal{Y})^{n_w} \to [0, 1]^k$, takes the features, $\mathbf{x}$, and generated potential outcomes, $\tilde{\mathbf{y}}$, and outputs a probability for each treatment, $w_1, ..., w_k$. Writing $\mathbf{D}_{\mathcal{W}}^w$ to denote the output of $\mathbf{D}_{\mathcal{W}}$ corresponding to treatment $w$, we define the loss, $\mathcal{L}_{\mathcal{W}}$, to be

$$\mathcal{L}_{\mathcal{W}}(\mathbf{D}_{\mathcal{W}}; \mathbf{G}) = -\mathbb{E}\left[ \sum_{w \in \mathcal{W}} \mathbb{I}_{\{W_f = w\}} \log \mathbf{D}_{\mathcal{W}}^w(\mathbf{X}, \tilde{\mathbf{Y}}) + \mathbb{I}_{\{W_f \neq w\}} \log(1 - \mathbf{D}_{\mathcal{W}}^w(\mathbf{X}, \tilde{\mathbf{Y}})) \right], \tag{9}$$

where, again, the expectation is taken over $\mathbf{X}, W_f, D_f, \tilde{\mathbf{Y}}$ and $\{\tilde{\mathcal{D}}_w\}_{w \in \mathcal{W}}$.

Then, for each $w \in \mathcal{W}$, $\mathbf{D}_w : \mathcal{X} \times (\mathcal{D}_w \times \mathcal{Y})^{n_w} \to [0, 1]^{n_w}$ is a map that takes the features, $\mathbf{x}$, and generated potential outcomes, $\tilde{y}_w$, corresponding to treatment $w$ and outputs a probability for each dosage level, $d_1^w, ..., d_{n_w}^w$, in a given realisation of $\tilde{\mathcal{D}}_w$. Writing $\mathbf{D}_w^j$ to denote the output of $\mathbf{D}_w$ corresponding to dosage level $D_j^w$, we define the loss of each dosage discriminator to be

$$\mathcal{L}_d(\mathbf{D}_w; \mathbf{G}) = -\mathbb{E}\left[ \mathbb{I}_{\{W_f = w\}} \sum_{j=1}^{n_w} \mathbb{I}_{\{D_f = D_j^w\}} \log \mathbf{D}_w^j(\mathbf{X}, \tilde{\mathbf{Y}}_w) + \mathbb{I}_{\{D_f \neq D_j^w\}} \log(1 - \mathbf{D}_w^j(\mathbf{X}, \tilde{\mathbf{Y}}_w)) \right], \tag{10}$$

where the expectation is taken over $\mathbf{X}, \tilde{\mathcal{D}}_w, \tilde{\mathbf{Y}}_w, W_f$ and $D_f$. The $\mathbb{I}_{\{W_f=w\}}$ term ensures that only samples for which the factual treatment is $w$ are used to train dosage discriminator $\mathbf{D}_w$ (otherwise there would be no factual dosage for that sample).

We define the overall discriminator $\mathbf{D}_H : \mathcal{X} \times \prod_{w \in \mathcal{W}} (\mathcal{D}_w \times Y)^{n_w} \to [0,1]^{\sum n_w}$ by defining its output corresponding to the treatment-dosage pair $(w, d_j^w)$ as

$$\mathbf{D}_H^{w,j}(\mathbf{x}, \tilde{\mathbf{y}}) = \mathbf{D}_\mathcal{W}^w(\mathbf{x}, \tilde{\mathbf{y}}) \times \mathbf{D}_w^j(\mathbf{x}, \tilde{\mathbf{y}}_w) . \tag{11}$$

Instead of the minimax game in Eq. 7, the generator and discriminator are trained according to the minimax game defined by seeking $\mathbf{G}^*, \mathbf{D}_H^*$ that solve

$$\mathbf{G}^* = \arg \min_{\mathbf{G}} \mathcal{L}(\mathbf{D}_H^*; \mathbf{G}) + \lambda \mathcal{L}_S(\mathbf{G}) \qquad \mathbf{D}_H^{*\,w,j} = \mathbf{D}_\mathcal{W}^{*\,w} \times \mathbf{D}_w^{*\,j}$$

$$\mathbf{D}_\mathcal{W}^* = \arg \min_{\mathbf{D}_\mathcal{W}} \mathcal{L}_\mathcal{W}(\mathbf{D}_\mathcal{W}; \mathbf{G}^*) \qquad \mathbf{D}_w^* = \arg \min_{\mathbf{D}_w} \mathcal{L}_d(\mathbf{D}_w; \mathbf{G}^*), \forall w \in \mathcal{W} \tag{12}$$

Fig. 2 depicts our generator and hierarchical discriminator. Pseudo-code for our algorithm can be found in Appendix C.

### 4.3 INFERENCE NETWORK

Once we have learned the counterfactual generator, we can use it only to access (generated) dose-response curves for all samples in the dataset. To generate dose-response curves for a new sample we use the counterfactual generator along with the original data to train an inference network, $\mathbf{I} : \mathcal{X} \times \mathcal{T} \to \mathcal{Y}$. Details of the loss and pseudo-code can be found in Appendix D.

## 5 ARCHITECTURE

In this section, we describe in detail the novel architectures that we adopt to model each of the functions $\mathbf{G}, \mathbf{D}, \mathbf{D}_\mathcal{W}, \mathbf{D}_{w_1}, ..., \mathbf{D}_{w_k}$ which draws from the ideas in Zaheer et al. (2017). The inference network, $\mathbf{I}$, has the same architecture as the generator, but does not receive $w_f, d_f, y_f$ or $\mathbf{z}$ as inputs.

### 5.1 GENERATOR ARCHITECTURE

We adopt a multi-task deep learning model for $\mathbf{G}$ by defining a function $g : \mathcal{X} \times \mathcal{T} \times \mathcal{Y} \times \mathcal{Z} \to \mathcal{H}$ for some latent space $\mathcal{H}$ (typically $\mathbb{R}^l$ for some $l$) and then for each treatment $w \in \mathcal{W}$ we introduce a multitask "head", $g_w : \mathcal{H} \times \mathcal{D}_w \to \mathcal{Y}$ taking inputs from $\mathcal{H}$ *and* a dosage, $d$, to produce an outcome $\hat{y}(w, d) \in \mathcal{Y}$. Given observations, $(\mathbf{x}, t_f, y_f)$, a noise vector $\mathbf{z}$, and a target treatment-dosage pair, $t = (w, d)$, we define

$$\mathbf{G}(\mathbf{x}, t_f, y_f, \mathbf{z})(t) = g_w(g(\mathbf{x}, t_f, y_f, \mathbf{z}), d) . \tag{13}$$

Each of $g, g_{w_1}, ..., g_{w_k}$ are modelled as fully connected networks. Fig. 3 depicts our generator architecture.

Figure 3: Generator architecture.

### 5.2 DISCRIMINATOR ARCHITECTURES

As noted in Section 1, our discriminators need to act as functions of sets (of randomly selected dosage-outcome pairs). While we could require that our discriminators try to learn this during training, by enforcing them to be functions of sets through their architecture, we reduce the complexity of learning the discriminators (they no longer need to "rule out" functions which are not functions of sets). This results in better performing discriminators, which in turn improves the performance of the generator.

In practice, the treatment discriminator receives all of the sets (i.e. one set for each treatment) of dosage-outcome pairs and outputs a probability for each treatment (i.e. there is one output corresponding to each set). In order to define such a function, we treat each input set as a vector but require that the outputs be invariant to (i.e. should not depend on) the ordering of the set as a vector.

Each dosage discriminator receives the set corresponding to a given treatment and is tasked with outputting a probability for each element in the set. In order to define such a function, we consider the input and output as vectors but then require that if we permute the elements of the input vector, the output should be permuted in the same way. We formalise the required notions - permutation invariance and permutation equivariance (Zaheer et al., 2017) - in the following subsection.

### 5.2.1 PERMUTATION INVARIANCE AND PERMUTATION EQUIVARIANCE

The notions of what it means for a function to be *permutation invariant* and *permutation equivariant* with respect to (a subset of) its inputs are given below in definitions 1 and 2, respectively. Let $\mathcal{U}, \mathcal{V}, \mathcal{C}$ be some spaces. Let $m \in \mathbb{Z}^+$.

**Definition 1.** *A function $f : \mathcal{U}^m \times \mathcal{V} \to \mathcal{C}$ is permutation invariant with respect to the space $\mathcal{U}^m$ if for every $\mathbf{u} = (u_1, ..., u_m) \in \mathcal{U}^m$, every $v \in \mathcal{V}$ and every permutation, $\sigma$, of $\{1, ..., m\}$ we have*

$$f(u_1, ..., u_m, v) = f(u_{\sigma(1)}, ..., u_{\sigma(m)}, v). \tag{14}$$

**Definition 2.** *A function $f : \mathcal{U}^m \times \mathcal{V} \to \mathcal{C}^m$ is permutation equivariant with respect to the space $\mathcal{U}^m$ if for every $\mathbf{u} = (u_1, ..., u_m) \in \mathcal{U}^m$, every $v \in \mathcal{V}$ and every permutation, $\sigma$, of $\{1, ..., m\}$ we have*

$$f(u_{\sigma(1)}, ..., u_{\sigma(m)}, v) = (f_{\sigma(1)}(\mathbf{u}, v), ..., f_{\sigma(m)}(\mathbf{u}, v)), \tag{15}$$

*where $f_j(\mathbf{u}, v)$ is the jth element of $f(\mathbf{u}, v)$.*

To build up functions that are permutation invariant and permutation equivariant we make the following observations: (1) the composition of any function with a permutation invariant function is permutation invariant, (2) the composition of two permutation equivariant functions is permutation equivariant.

Zaheer et al. (2017) provide several possible building blocks to use to construct invariant and equivariant deep networks. The basic building block we will use for invariant functions will be a layer of the form

$$f_{inv}(\mathbf{u}) = \sigma(\mathbf{1}_b \mathbf{1}_m^T (\phi(u_1), ..., \phi(u_m))), \tag{16}$$

where $\mathbf{1}_l$ is a vector of 1s of dimension $l$, $\phi$ is any function $\phi : \mathcal{U} \to \mathbb{R}^q$ for some $q$ (in this paper we use a standard fully connected layer) and $\sigma$ is some non-linearity.

The basic building block for equivariant functions is defined in terms of an equivariance input, $\mathbf{u}$, and an auxiliary input, $\mathbf{v}$, by

$$f_{equi}(\mathbf{u}, \mathbf{v}) = \sigma(\lambda \mathbf{I}_m \mathbf{u} + \gamma(\mathbf{1}_m \mathbf{1}_m^T)\mathbf{u} + (\mathbf{1}_m \Theta^T)\mathbf{v}), \tag{17}$$

where $\mathbf{I}_m$ is the $m \times m$ identity matrix, $\lambda$ and $\gamma$ are scalar parameters and $\Theta$ is a vector of weights.

### 5.2.2 HIERARCHICAL DISCRIMINATOR ARCHITECTURE

In the case of the hierarchical discriminator, we want the treatment discriminator, $\mathbf{D}_{\mathcal{W}}$, to be permutation invariant with respect to $\tilde{\mathbf{y}}_w$ for each treatment, $w \in \mathcal{W}$. To achieve this we define a function $h_1 : \prod_{w \in \mathcal{W}} (\mathcal{D}_w \times \mathcal{Y})^{n_w} \to \mathcal{H}_H$ and require that this function be permutation invariant with respect to each of the spaces $(\mathcal{D}_w \times \mathcal{Y})^{n_w}$. We then concatenate the output of $h_1$ with the features $\mathbf{x}$ and pass these through a fully connected network $h_2 : \mathcal{X} \times \mathcal{H}_H \to [0, 1]^k$ so that

$$\mathbf{D}_{\mathcal{W}}(\mathbf{x}, \tilde{\mathbf{y}}) = h_2(\mathbf{x}, h_1(\tilde{\mathbf{y}})). \tag{18}$$

To construct $h_1$, we concatenate the outputs of several invariant layers of

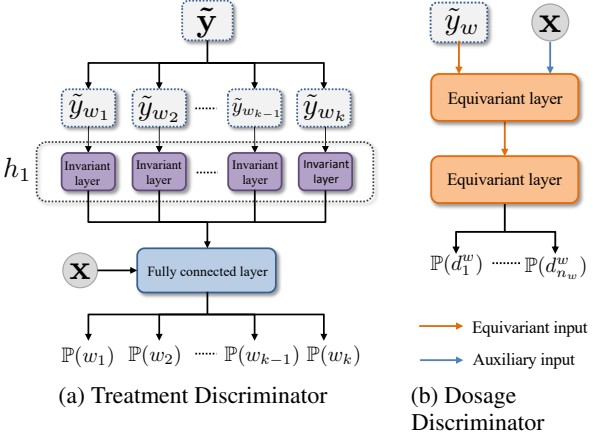

(a) Treatment Discriminator

(b) Dosage Discriminator

Figure 4: Architecture of our discriminators.

the form given in Eq. (16) that each individually act on the spaces $(\mathcal{D}_w \times \mathcal{Y})^{n_w}$. That is, for each treatment, $w \in \mathcal{W}$ we define a map $h^w_{inv} : (\mathcal{D}_w \times \mathcal{Y})^{n_w} \to \mathcal{H}^w_H$ by substituting $\tilde{\mathbf{y}}_w$ for $\mathbf{u}$ in Eq. (16). We then define $\mathcal{H}_H = \prod_{w \in \mathcal{W}} \mathcal{H}^w_H$ and $h_1(\tilde{\mathbf{y}}) = (h^{w_1}_{inv}(\tilde{\mathbf{y}}_{w_1}), ..., h^{w_k}_{inv}(\tilde{\mathbf{y}}_{w_k}))$.

We want each dosage discriminator, $\mathbf{D}_w$, to be permutation equivariant with respect to $\tilde{\mathbf{y}}_w$. To achieve this each $\mathbf{D}_w$ will consist of two layers of the form given in Eq. (17) with the equivariance input, $\mathbf{u}$, to the first layer being $\tilde{\mathbf{y}}_w$ and to the second layer being the output of the first layer and the auxiliary input, $\mathbf{v}$, to the first layer being the features, $\mathbf{x}$, and then no auxiliary input to the second layer.

Diagrams depicting the architectures of the treatment discriminator and dosage discriminators can be found in Fig. 4(a) and Fig. 4(b) respectively.

# 6 EVALUATION

The nature of the treatment-effects estimation problem in even the binary treatments setting does not allow for meaningful evaluation on real-world datasets. While there are well-established benchmark synthetic models for use in the binary (or multiple) case, no such models exist for the dosage setting. We propose our own semi-synthetic data simulation to evaluate our model against several benchmarks.

## 6.1 EXPERIMENTAL SETUP

**Semi-synthetic data generation:** We simulate data as follows. We obtain features, $\mathbf{x}$, from a real dataset (in this paper we use TCGA (Weinstein et al., 2013), News (Johansson et al., 2016; Schwab et al., 2019)) and MIMIC III (Johnson et al., 2016))[3]. We consider 3 treatments each accompanied by a dosage. Each treatment, $w \in \mathcal{W}$, is associated with a set of parameters, $\mathbf{v}^w_1, \mathbf{v}^w_2, \mathbf{v}^w_3$. For each run of the experiment, these parameters are sampled randomly by first sampling a vector, $\mathbf{u}^w_i$, from $\mathcal{N}(\mathbf{0}, \mathbf{1})$ and then setting $\mathbf{v}^w_i = \mathbf{u}^w_i / ||\mathbf{u}^w_i||$ where $|| \cdot ||$ is the standard Euclidean norm. The shape of the dose-response curve for each treatment, $f_w(\mathbf{x}, d)$, is given in Table 1, along with a closed-form expression for the optimal dosage. We add $\epsilon \sim \mathcal{N}(0, 0.2)$ noise to the outcomes.

We assign factual treatment-dosage pairs to each sample by first sampling a dosage, $d_w$, for each treatment from a beta distribution, $d_w | \mathbf{x} \sim \text{Beta}(\alpha, \beta_w)$. The parameter $\alpha \geq 1$ controls the dosage selection bias[4] and the parameter $\beta_w$ is set to $\beta_w = \frac{\alpha-1}{d^*_w} + 2 - \alpha$, with $d^*_w$ being the optimal dosage for each treatment[5]. This setting of $\beta_w$ ensures that the mode of the Beta distribution is the optimal dosage. Once we have sampled a dosage for each treatment, we assign a treatment according to $w_f | \mathbf{x} \sim \text{Categorical}(\text{softmax}(\kappa f(\mathbf{x}, d_w))$ where a higher $\kappa$ will result in a stronger selection bias, and $\kappa = 0$ results in the treatments being assigned completely randomly. The factual treatment-dosage pair is then given by $(w_f, d_{w_f})$. Unless otherwise specified, we set $\kappa = 2$ and $\alpha = 2$.

We consider 3 different shapes for $f_w$ to demonstrate learning heterogeneous dose-response curves. The first curve can be broken down into two terms, a linear (in $d$) increasing term $(\mathbf{v}^1_1)^T\mathbf{x} + 12(\mathbf{v}^1_2)^T\mathbf{x}d$ and a quadratic (in $d$) decreasing term $-12(\mathbf{v}^1_3)^T\mathbf{x}d^2$. This first term could, for example, represent the improved efficacy of higher dosages of chemotherapy in reducing the size of a tumour, while the quadratic term could represent the increasing toxicity of chemotherapy as the dosage increases. This type of trade-off presents itself in many other settings where there are both costs and rewards.

For metrics, we use Mean Integrated Square Error (MISE), Dosage Policy Error (DPE) and Policy Error (PE) (Silva, 2016; Schwab et al., 2019). Details can be found in Appendix F.

**Benchmarks:** We compare against two benchmarks: Generalized Propensity Score (GPS) (Imbens, 2000) and Dose Reponse Networks (DRNet) (Schwab et al., 2019). For DRNets, we compare against both the standard model architecture described by Schwab et al. (2019) as well as with Wasserstein regularization (DRN-W).

---

[3]Details of each dataset along with suggested interpretations of the synthetic treatments and outcomes for each dataset can be found in Appendix G

[4]When $\alpha = 1$, $\beta_w = 1$ and $\text{Beta}(\alpha, \beta_w)$ reduces to the uniform distribution. See Appendix H.

[5]If the optimal dosage is 0, we sample $d_w$ from $1 - \text{Beta}(\alpha, \beta_w)$ where $\beta_w$ is set as though $d^*_w = 1$. This results in the dosage being sampled symmetrically for $d^*_w = 0$ and $d^*_w = 1$.

| Treatment | Dose-Response | Optimal dosage |
|---|---|---|
| 1 | $f_1(\mathbf{x}, d) = C((\mathbf{v}_1^1)^T\mathbf{x} + 12(\mathbf{v}_2^1)^T\mathbf{x}d - 12(\mathbf{v}_3^1)^T\mathbf{x}d^2)$ | $d_1^* = \frac{(\mathbf{v}_2^1)^T\mathbf{x}}{2(\mathbf{v}_3^1)^T\mathbf{x}}$ |
| 2 | $f_2(\mathbf{x}, d) = C((\mathbf{v}_1^2)^T\mathbf{x} + \sin(\pi(\frac{\mathbf{v}_2^2{}^T\mathbf{x}}{\mathbf{v}_3^2{}^T\mathbf{x}})d))$ | $d_2^* = \frac{(\mathbf{v}_3^2)^T\mathbf{x}}{2(\mathbf{v}_2^2)^T\mathbf{x}}$ |
| 3 | $f_3(\mathbf{x}, d) = C((\mathbf{v}_1^3)^T\mathbf{x} + 12d(d-b)^2$, where $b = 0.75\frac{(\mathbf{v}_2^3)^T\mathbf{x}}{(\mathbf{v}_3^3)^T\mathbf{x}})$ | $\frac{b}{3}$ if $b \geq 0.75$ 
 1 if $b < 0.75$ |

Table 1: Dose response curves used to generate semi-synthetic outcomes for patient features $\mathbf{x}$. In the experiments, we set $C = 10$. $\mathbf{v}_1^w, \mathbf{v}_2^w, \mathbf{v}_3^w$ are the parameters associated with each treatment $w$.

As a baseline for comparison, we also use a standard multilayer perceptron (MLP) that takes as input the patient features, the treatment and dosage and estimates the patient outcome and a multitask variant (MLP-M) that has a designated head for each treatment. See Appendix E for details of the benchmark models and their hyperparameter optimisation.

## 6.2 SOURCE OF GAIN

Before comparing against the benchmarks, we investigate how each component of our model affects performance. We start with a baseline model in which both the generator and discriminator consist of a single fully connected network. One at a time, we add in the following components (cumulatively until we reach our full model): (1) the supervised loss in Eq. 4 (+ $\mathcal{L}_S$), (2) multitask heads in the generator (+ Multitask), (3) hierarchical discriminator (+ Hierarchical) and (4) invariance/equivariance layers in the treatment and dosage discriminators (+Inv/Eqv). We report the results in Table 2 for TCGA and News for all 3 error metrics (MISE, DPE and PE), computed over 30 runs.

| | TCGA | | | News | | |
|---|---|---|---|---|---|---|
| | $\sqrt{\text{MISE}}$ | $\sqrt{\text{DPE}}$ | $\sqrt{\text{PE}}$ | $\sqrt{\text{MISE}}$ | $\sqrt{\text{DPE}}$ | $\sqrt{\text{PE}}$ |
| Baseline | $4.18 \pm 0.32$ | $2.06 \pm 0.16$ | $1.93 \pm 0.12$ | $6.17 \pm 0.27$ | $6.97 \pm 0.27$ | $6.20 \pm 0.21$ |
| + $\mathcal{L}_S$ | $3.37 \pm 0.11$ | $1.14 \pm 0.05$ | $0.84 \pm 0.05$ | $4.51 \pm 0.16$ | $4.46 \pm 0.12$ | $4.40 \pm 0.11$ |
| + Multitask | $3.15 \pm 0.12$ | $0.85 \pm 0.05$ | $0.67 \pm 0.05$ | $4.11 \pm 0.11$ | $4.33 \pm 0.11$ | $4.31 \pm 0.11$ |
| + Hierarchical | $2.54 \pm 0.05$ | $0.36 \pm 0.05$ | $0.45 \pm 0.05$ | $4.07 \pm 0.05$ | $4.24 \pm 0.11$ | $4.17 \pm 0.12$ |
| + Inv/Eqv | $1.89 \pm 0.05$ | $0.31 \pm 0.05$ | $0.25 \pm 0.05$ | $3.71 \pm 0.05$ | $4.14 \pm 0.11$ | $3.90 \pm 0.05$ |

Table 2: Source of gain analysis for our model. Metrics are reported as Mean $\pm$ Std.

We see that the addition of each component results in a performance improvement for our model, with the final row (which corresponds to our full model) demonstrating the best performance across both datasets and for all metrics.

To further demonstrate the advantages of our hierarchical discriminator, in Fig. 5 we investigate how our hierarchical discriminator compares with a single network discriminator (all other components are included in both models, see Appendix B for details of the single discriminator) when we vary the hyperparameter $n_w$ on TCGA. Similar results for News can be found in Appendix I.1.

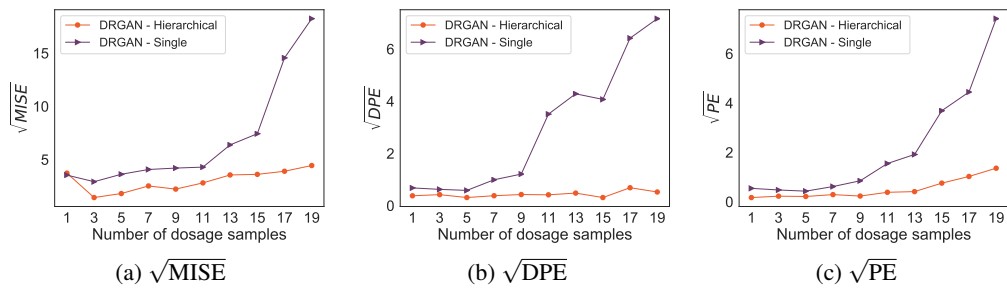

(a) $\sqrt{\text{MISE}}$       (b) $\sqrt{\text{DPE}}$       (c) $\sqrt{\text{PE}}$

Figure 5: Performance of single vs. hierarchical discriminator when increasing the number of dosage samples ($n_w$) on TCGA dataset.

The performance of the single discriminator causes significant performance drops around $n_w = 9$ across all metrics. As previously noted, this is due to the dimension of the output space (which for $n_w = 9$ is 27) being too large. Conversely, we see that our hierarchical discriminator shows much more stable performance even when $n_w = 19$. We investigate in Appendix I.1 the hyperparameter $\lambda$.

## 6.3 BENCHMARKS COMPARISON

We now compare DRGAN against the benchmarks on our 3 semi-synthetic datasets. For Mimic, due to the low number of samples available, we use only two treatments - 2 and 3. We report $\sqrt{\text{MISE}}$ and $\sqrt{\text{PE}}$ in Table 3, with results for $\sqrt{\text{DPE}}$ given in Appendix I.3. We see that DRGAN demonstrates a statistically significant improvement over every benchmark across all 3 datasets, confirming that DRGAN is able to learn response-curves on top of very different underlying patient features.

| Method | TCGA | | News | | MIMIC | |
|--------|------|------|------|------|-------|------|
| | $\sqrt{\text{MISE}}$ | $\sqrt{\text{PE}}$ | $\sqrt{\text{MISE}}$ | $\sqrt{\text{PE}}$ | $\sqrt{\text{MISE}}$ | $\sqrt{\text{PE}}$ |
| DRGAN | $\mathbf{1.89 \pm 0.05}$ | $\mathbf{0.25 \pm 0.05}$ | $\mathbf{3.71 \pm 0.05}$ | $\mathbf{3.90 \pm 0.05}$ | $\mathbf{2.09 \pm 0.12}$ | $\mathbf{0.32 \pm 0.05}$ |
| DRNet | $3.64 \pm 0.12$ | $0.67 \pm 0.05$ | $4.98 \pm 0.12$ | $4.17 \pm 0.11$ | $4.45 \pm 0.12$ | $1.44 \pm 0.05$ |
| DRN-W | $3.71 \pm 0.12$ | $0.63 \pm 0.05$ | $5.07 \pm 0.12$ | $4.56 \pm 0.12$ | $4.47 \pm 0.12$ | $1.37 \pm 0.05$ |
| GPS | $4.83 \pm 0.01$ | $1.60 \pm 0.01$ | $6.97 \pm 0.01$ | $24.1 \pm 0.05$ | $7.39 \pm 0.00$ | $20.2 \pm 0.01$ |
| MLP-M | $3.96 \pm 0.12$ | $1.20 \pm 0.05$ | $5.17 \pm 0.12$ | $5.82 \pm 0.16$ | $4.97 \pm 0.16$ | $1.59 \pm 0.05$ |
| MLP | $4.31 \pm 0.05$ | $0.97 \pm 0.05$ | $5.48 \pm 0.16$ | $6.45 \pm 0.21$ | $5.34 \pm 0.16$ | $1.65 \pm 0.05$ |

Table 3: Performance of individualized treatment-dose response estimation on three datasets. Bold indicates the method with the best performance for each dataset.

In Appendix I.4 we compare DRGAN with DRNET and GPS for an increasing number of treatments.

## 6.4 TREATMENT AND DOSAGE SELECTION BIAS

In this section, we assess the robustness of each method to varying treatment and dosage bias. We report results for $\sqrt{\text{MISE}}$ on TCGA here. For the other metrics see Appendix I.2. Fig. 6(a) shows the performance of the 4 methods for $\kappa$ between 0 (no bias) and 10 (strong bias). Fig. 6(b) shows the performance for $\alpha$ between 1 (no bias) and 8 (strong bias). We see that our model shows consistent performance, significantly outperforming the benchmark methods across the entire ranges of $\kappa$ and $\alpha$.

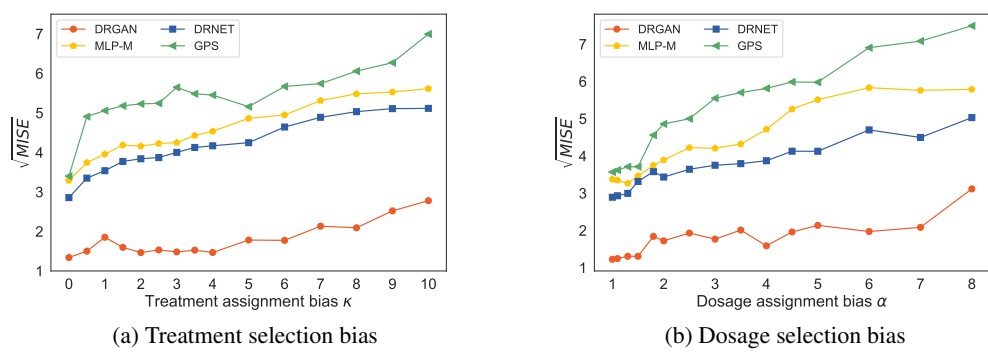

(a) Treatment selection bias       (b) Dosage selection bias

Figure 6: Performance of the 4 methods on datasets with varying bias levels.

## 7 CONCLUSION

In this paper we proposed a novel framework for estimating dose-response curves from observational data. Our method modified the GAN framework, introducing a novel hierarchical discriminator for use in the dose-response setting. We also proposed novel architectures for the networks involved in our model and introduced a new semi-synthetic data simulation for use as a benchmark in this setting. On this data we demonstrated significant improvements over the benchmarks.

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

APPENDIX

# A    EXPANDED RELATED WORKS

Most methods for performing causal inference in the static setting focus on the scenario with two or multiple treatment options and no dosage parameter. The approaches taken by such methods to estimate the treatment effects involve either building a separate regression model for each treatment (Stoehlmacher et al., 2004; Qian & Murphy, 2011; Bertsimas et al., 2017) or using the treatment as a feature and adjusting for the imbalance between the different treatment populations. The former does not generalise to the dosage setting due to the now infinite number of possible treatments available. In the latter case, methods for handling the selection bias involve propensity weighting (Crump et al., 2008; Alaa et al., 2017; Shi et al., 2019), building sub-populations using tree based methods (Chipman et al., 2010; Athey & Imbens, 2016; Wager & Athey, 2018; Kallus, 2017) or building balancing representations between patients receiving the different treatments (Johansson et al., 2016; Shalit et al., 2017; Li & Fu, 2017; Yao et al., 2018). An additional approach involves modelling the data distribution of the factual and counterfactual outcomes (Alaa & van der Schaar, 2017; Yoon et al., 2018).

Silva (2016) leverages observational and interventional data to estimate the effects of discrete dosages for a single treatment. In particular, Silva (2016) uses observational data to construct a non-stationary covariance function and develop a hierarchical Gaussian process prior to build a distribution over the dose response curve. Then, controlled interventions are employed to learn a non-parametric affine transform to reshape this distribution. The setting in Silva (2016) differs significantly from ours as we do not assume access to any interventional data.

## B  SINGLE DISCRIMINATOR MODEL

In the paper we developed a hierarchical discriminator and demonstrated that it performs significantly better than the single discriminator setup that we now describe in this section.

### B.1  SINGLE DISCRIMINATOR

In the single model, we will aim to learn a single discriminator, $\mathbf{D}$, that outputs $\mathbb{P}((W_f, D_f) = (w, d)|\mathbf{X}, \tilde{\mathcal{D}}_w, \tilde{\mathbf{Y}})$ for each $w \in \mathcal{W}$ and $d \in \tilde{\mathcal{D}}_w$. We will write $\mathbf{D}^{w,d}(\cdot)$ to denote the output of $\mathbf{D}$ that corresponds to the treatment-dosage pair $(w, d)$. We define the loss, $\mathcal{L}_D$, to be

$$\mathcal{L}_D(\mathbf{D}; \mathbf{G}) = -\mathbb{E}\left[ \sum_{w \in \mathcal{W}} \sum_{d \in \tilde{\mathcal{D}}_w} \mathbb{I}_{\{T_f=(w,d)\}} \log \mathbf{D}^{w,d}(\mathbf{X}, \tilde{\mathbf{Y}}) + \mathbb{I}_{\{T_f \neq (w,d)\}} \log(1 - \mathbf{D}^{w,d}(\mathbf{X}, \tilde{\mathbf{Y}})) \right] \tag{19}$$

where the expectation is taken over $\mathbf{X}, \{\tilde{\mathcal{D}}_w\}_{w \in \mathcal{W}}, \tilde{\mathbf{Y}}, W_f$ and $D_f$ and we note that the dependence on $\mathbf{G}$ is through $\tilde{\mathbf{Y}}$. Our single discriminator will be trained to minimise this loss directly. The generator GAN-loss, $\mathcal{L}_G$, is then defined by

$$\mathcal{L}_G(\mathbf{G}) = -\mathcal{L}_D(\mathbf{D}^*; \mathbf{G}) \tag{20}$$

where $\mathbf{D}^*$ is the optimal discriminator given by minimising $\mathcal{L}_D$. The generator will be trained to minimise $\mathcal{L}_G + \lambda\mathcal{L}_S$.

### B.2  SINGLE DISCRIMINATOR ARCHITECTURE

In the case of the single discriminator, we want the output of $\mathbf{D}$ corresponding to each treatment $w \in \mathcal{W}$, i.e. $(\mathbf{D}^{w,1}, ..., \mathbf{D}^{w,n_w})$, to be permutation equivariant with respect to $\tilde{\mathbf{y}}_w$ and permutation invariant with respect to each $\tilde{\mathbf{y}}_v$ for $v \in \mathcal{W} \backslash \{w\}$. To achieve this, we first define a function $f : \prod_{w \in \mathcal{W}}(\mathcal{D}_w \times \mathcal{Y})^{n_w} \to \mathcal{H}_S$ and require that this function be permutation invariant with respect to each of the spaces $(\mathcal{D}_w \times \mathcal{Y})^{n_w}$. For each treatment, $w \in \mathcal{W}$, we introduce a multi-task head, $f_w : \mathcal{X} \times \mathcal{H}_S \times (\mathcal{D}_w \times \mathcal{Y})^{n_w} \to [0, 1]^{n_w}$, and require that each of these functions be permutation equivariant with respect to their corresponding input space $(\mathcal{D}_w \times \mathcal{Y})^{n_w}$ but they can depend on the features, $\mathbf{x} \in \mathcal{X}$, and invariant latent representation coming from $f$ arbitrarily. Writing $f_w^j$ to denote the $j$th output of $f_w$, the output of the discriminator given input features, $\mathbf{x}$, and generated outcomes, $\tilde{\mathbf{y}}$, is defined by

$$\mathbf{D}^{w,j}(\mathbf{x}, \tilde{\mathbf{y}}) = f_w^i(\mathbf{x}, f(\tilde{\mathbf{y}}), \tilde{\mathbf{y}}_w). \tag{21}$$

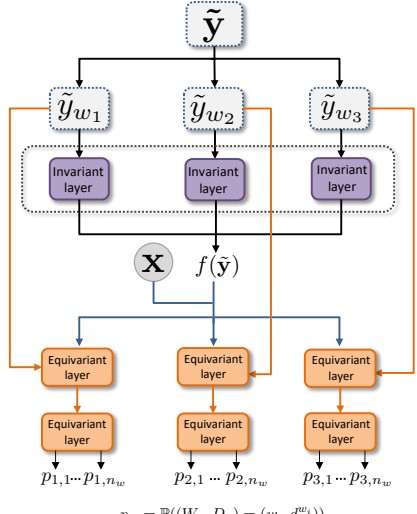

Figure 7: Overview of the single discriminator architecture.

To construct the function $f$, we concatenate the outputs of several invariant layers of the form given in Eq. (16) that each individually act on the spaces $(\mathcal{D}_w \times \mathcal{Y})^{n_w}$. That is, for each treatment, $w \in \mathcal{W}$ we define a map $f_{inv}^w : (\mathcal{D}_w \times \mathcal{Y})^{n_w} \to \mathcal{H}_S^w$ by substituting $\tilde{\mathbf{y}}_w$ for $\mathbf{u}$ in Eq. (16). We then define $\mathcal{H}_S = \prod_{w \in \mathcal{W}} \mathcal{H}_S^w$ and $f(\tilde{\mathbf{y}}) = (f_{inv}^{w_1}(\tilde{\mathbf{y}}_{w_1}), ..., f_{inv}^{w_k}(\tilde{\mathbf{y}}_{w_k}))$.

Each $f_w$ will consist of two layers of the form given in Eq. (17) with the equivariance input, $\mathbf{u}$, to first layer being $\tilde{\mathbf{y}}_w$ and to the second layer being the output of the first layer and the auxiliary input, $\mathbf{v}$, to the first layer being the concatenation of the features and invariant representation, i.e. $(\mathbf{x}, f(\tilde{\mathbf{y}}))$ and then no auxiliary input to the second layer.

A diagram depicting the architecture of the single discriminator model can be found in Fig. 7.

## C    COUNTERFACTUAL GENERATOR PSEUDO-CODE

---

**Algorithm 1** Training of the generator in DRGAN

---

1: **Input:** dataset $\mathcal{C} = \{(\mathbf{x}^i, t^i_f, y^i_f) : i = 1, ..., N\}$, batch size $n_{mb}$, number of dosages per treatment $n_d$, number of discriminator updates per iteration $n_D$, number of generator updates per iteration $n_G$, dimensionality of noise $n_z$, learning rate $\alpha$
2: **Initialize:** $\theta_G, \theta_{\mathcal{W}}, \{\theta_w\}_{w \in \mathcal{W}}$
3: **while G** has not converged **do**
   Discriminator updates
4:     **for** $i = 1, ..., n_D$ **do**
5:         Sample $(\mathbf{x}_1, (w_1, d_1), y_1), ..., (\mathbf{x}_{n_{mb}}, (w_{n_{mb}}, d_{n_{mb}}), y_{n_{mb}})$ from $\mathcal{C}$
6:         Sample generator noise $\mathbf{z}_j = (z^j_1, ..., z^j_{n_z})$ from $\text{Unif}([0, 1]^{n_z})$ for $j = 1, ..., n_{mb}$
7:         **for** $w \in \mathcal{W}$ **do**
8:             **for** $j = 1, ..., n_{mb}$ **do**
9:                 Sample $\tilde{D}^j_w = (d^{w,j}_1, ..., d^{w,j}_{n_d})$ independently and uniformly from $(\mathcal{D}_w)^{n_d}$
10:                Set $\tilde{\mathbf{y}}^j_w$ according to Eq. 5
11:                Calculate gradient of dosage discriminator loss

$$g_w \leftarrow \nabla_{\theta_w} - \left[ \sum_{\{j : w_j = w\}} \sum_{k=1}^{n_d} \mathbb{I}_{\{d_j = d^{w,j}_k\}} \log \mathbf{D}_w(\mathbf{x}_j, \tilde{\mathbf{y}}^j_w) + \mathbb{I}_{\{d_j \neq d^{w,j}_k\}} \log(1 - \mathbf{D}_w(\mathbf{x}_j, \tilde{\mathbf{y}}^j_w)) \right]$$

12:                Update dosage discriminator parameters $\theta_w \leftarrow \theta_w + \alpha g_w$
13:            Set $\tilde{\mathbf{y}}_j = (\tilde{\mathbf{y}}^j_w)_{w \in \mathcal{W}}$
14:            Calculate gradient of treatment discriminator loss

$$g_{\mathcal{W}} \leftarrow \nabla_{\theta_{\mathcal{W}}} - \left[ \sum_{j=1}^{n_{mb}} \sum_{w \in \mathcal{W}} \mathbb{I}_{\{w_j = w\}} \log \mathbf{D}_{\mathcal{W}}(\mathbf{x}_j, \tilde{\mathbf{y}}_j) + \mathbb{I}_{\{w_j \neq w\}} \log(1 - \mathbf{D}_{\mathcal{W}}(\mathbf{x}_j, \tilde{\mathbf{y}}_j)) \right]$$

15:            Update treatment discriminator parameters $\theta_{\mathcal{W}} \leftarrow \theta_{\mathcal{W}} + \alpha g_{\mathcal{W}}$
   Generator updates
16:    **for** $i = 1, ..., n_G$ **do**
17:        Sample $(\mathbf{x}_1, (w_1, d_1), y_1), ..., (\mathbf{x}_{n_{mb}}, (w_{n_{mb}}, d_{n_{mb}}), y_{n_{mb}})$ from $\mathcal{C}$
18:        Sample generator noise $\mathbf{z}_j = (z^j_1, ..., z^j_{n_z})$ from $\text{Unif}([0, 1]^{n_z})$ for $j = 1, ..., n_{mb}$
19:        Sample $(\tilde{D}^j_w)_{w \in \mathcal{W}}$ from $\Pi_{w \in \mathcal{W}} (\mathcal{D}_w)^{n_d}$ for $j = 1, ..., n_{mb}$
20:        Set $\tilde{\mathbf{y}}$ according to Eq. 5
21:        Calculate gradient of generator loss

$$g_G \leftarrow \nabla_{\theta_G} \left[ \sum_{j=1}^{n_{mb}} \sum_{w \in \mathcal{W}} \sum_{l=1}^{n_d} \mathbb{I}_{\{w_j = w, d_j = d^{w,j}_l\}} \log(\mathbf{D}^w_{\mathcal{W}}(\mathbf{x}_j, \tilde{\mathbf{y}}_j)_w \times \mathbf{D}^l_w(\mathbf{x}_j, \tilde{\mathbf{y}}^j_w)_l) \right.$$

$$\left. + \mathbb{I}_{\{w_j \neq w, d_j \neq d^{w,j}_l\}} \log(1 - (\mathbf{D}^w_{\mathcal{W}}(\mathbf{x}_j, \tilde{\mathbf{y}}_j) \times D^l_w(\mathbf{x}_j, \tilde{\mathbf{y}}^j_w))) \right]$$

22:        Update generator parameters $\theta_G \leftarrow \theta_G + \alpha g_G$
23: **Output: G**

---

## D INFERENCE NETWORK

To generate dose-response curves for new samples, we learn an inference network, $\mathbf{I} : \mathbf{X} \times \mathcal{T} \to \mathcal{Y}$. This inference network is trained using the original dataset and the learned counterfactual generator. As with the training of the generator and discriminator, we train using a random set of dosages, $\tilde{\mathcal{D}}_w$. The loss is given by

$$\mathcal{L}_I(\mathbf{I}) = \mathbb{E}\left[ \sum_{w \in \mathcal{W}} \sum_{d \in \tilde{\mathcal{D}}_w} (\tilde{Y}(w, d) - \mathbf{I}(\mathbf{X}, (w, d)))^2 \right], \tag{22}$$

where $\tilde{Y}(w, d)$ is $Y_f$ if $T_f = (w, d)$ or given by the generator if $T_f \neq (w, d)$. The expectation is taken over $\mathbf{X}, T_f, Y_f, \mathbf{Z}$ and $\tilde{\mathcal{D}}_w$.

### D.1 PSEUDO-CODE FOR TRAINING THE INFERENCE NETWORK

---
**Algorithm 2** Training of the inference network in DRGAN
---
1: **Input:** dataset $\mathcal{C} = \{(\mathbf{x}^i, t_f^i, y_f^i) : i = 1, ..., N\}$, trained generator $\mathbf{G}$, batch size $n_{mb}$, number of dosages per treatment $n_d$, dimensionality of noise $n_z$, learning rate $\alpha$
2: **Initialize:** $\theta_I$,
3: **while I** has not converged **do**
4:     Sample $(\mathbf{x}_1, (w_1, d_1), y_1), ..., (\mathbf{x}_{n_{mb}}, (w_{n_{mb}}, d_{n_{mb}}), y_{n_{mb}})$ from $\mathcal{C}$
5:     Sample generator noise $\mathbf{z}_j = (z_1^j, ..., z_{n_z}^j)$ from $\text{Unif}([0, 1]^{n_z})$ for $j = 1, ..., n_{mb}$
6:     **for** $j = 1, ..., n_{mb}$ **do**
7:         **for** $w \in \mathcal{W}$ **do**
8:             Sample $\tilde{D}_w^j = (d_1^{w,j}, ..., d_{n_d}^{w,j})$ independently and uniformly from $(\mathcal{D}_w)^{n_d}$
9:             Set $\tilde{\mathbf{y}}_w^j$ according to Eq. 5 5
10:     Calculate gradient of inference network loss

$$g_I \leftarrow \nabla_{\theta_I}\left[ \sum_{j=1}^{n_{mb}} \sum_{w \in \mathcal{W}} \sum_{l=1}^{n_d} (\tilde{\mathbf{y}}_w^j)_l - \mathbf{I}(\mathbf{x}_j, (w, d_l^{w,j}))^2 \right]$$

11:     Update inference network parameters $\theta_I \leftarrow \theta_I + \alpha g_I$
12: **Output: I**

---

# E  BENCHMARKS

We use the publicly available GitHub implementation of DRNet provided by Schwab et al. (2019): `https://github.com/d909b/drnet`. Moreover, we also used a GPS implementation similar to the one from `https://github.com/d909b/drnet` which uses the `causaldrf` R package (Galagate, 2016). More spcifically, the GPS implementation uses a normal treatment model, a linear treatment formula and a 2-nd degree polynomial for the outcome. Moreover, for the TCGA and News datasets, we performed PCA and only used the 50 principal components as input to the GPS model to reduce computational complexity.

**Hyperparameter optimization:** The validation split of the dataset is used for hyperparameter optimization. For the DRNet benchmarks we use the same hyperparameter optimization proposed by Schwab et al. (2019) with the hyperparameter search ranges described in Table 4. For DRGAN, we use the hyperparameter optimization method proposed in GANITE (Yoon et al., 2018), where we use the complete dataset from the counterfactual generator to evaluate the MISE on the inference network. We perform a random search (Bergstra & Bengio, 2012) for hyperparameter optimization over the search ranges in Table 5. For a fair comparison, for the MLP-M model we used the same architecture used in the inference network of DRGAN. Similarly, for the MLP model we use the same architecture as for the MLP-M, but without the multitask heads.

| Hyperparameter | Search range |
|---|---|
| Batch size | 32, 64, 128 |
| Number of units per hidden layer | 24, 48, 96, 192 |
| Number of hidden layers | 2, 3 |
| Dropout percentage | 0.0, 0.2 |
| Imbalance penalty weight* | 0.1, 1.0, 10.0 |
| | Fixed |
| Number of dosage strata $E$ | 5 |

Table 4: Hyperparameters search range for DRNet. *: For the DRNet model using Wasserstein regularization only.

| Hyperparameter | Search range |
|---|---|
| Batch size | 64, 128, 256 |
| Number of units per hidden layer | 32, 64, 128 |
| Size of invariant and equivariant representations | 16, 32, 64, 128 |
| | Fixed |
| Number of hidden layers per multitask head | 2 |
| Number of dosage samples | 5 |
| $\lambda$ | 1 |
| Optimization | Adam Moment Optimization |

Table 5: Hyperparameters search range for DRGAN.

## F    METRICS

The Mean Integrated Square Error (MISE) measures how well the models estimates the patient outcome across the entire dosage space:

$$\text{MISE} = \frac{1}{N}\frac{1}{k}\sum_{w\in\mathcal{W}}\sum_{i=1}^{N}\int_{\mathcal{D}_w}\left(y^i(w,u) - \hat{y}^i(w,u)\right)^2 du\,. \tag{23}$$

In addition to this, we also compute the mean dosage policy error (DPE) (Schwab et al., 2019) to assess the ability of the model to estimate the optimal dosage point for every treatment for each individual:

$$\text{DPE} = \frac{1}{N}\frac{1}{k}\sum_{w\in\mathcal{W}}\sum_{i=1}^{N}\left(y^i(w,d_w^*) - y^i(w,\hat{d}_w^*)\right)^2\,, \tag{24}$$

where $d_w^*$ is the true optimal dosage and $\hat{d}_w^*$ is the optimal dosage identified by the model. The optimal dosage points for a model are computed using SciPy's implementation of Sequential Least SQuares Programming.

Finally, we compute the mean policy error (PE) (Schwab et al., 2019) which compares the outcome of the true optimal treatment-dosage pair to the outcome of the optimal treatment-dosage pair as selected by the model:

$$\text{PE} = \frac{1}{N}\sum_{i=1}^{N}\left(y^i(w^*,d_w^*) - y^i(\hat{w}^*,\hat{d}_w^*)\right)^2\,, \tag{25}$$

where $w^*$ is the true optimal treatment and $\hat{w}^*$ is the optimal treatment identified by the model. The optimal treatment-dosage pair for a model is selected by first computing the optimal dosage for each treatment and then selecting the treatment with the best outcome for its optimal dosage.

Each of these metrics are computed on a held out test-set.

## G  DATASET DESCRIPTIONS

**TCGA:** The TCGA dataset consists of gene expression measurements for cancer patients (Weinstein et al., 2013). There are 9659 samples for which we used the measurements from the 4000 most variable genes. The gene expression data was log-normalized and each feature was scaled in the $[0, 1]$ interval. Moreover, for each patient, the features $\mathbf{x}$ were scaled to have norm 1. We give meaning to our treatments and dosages by considering the treatment as being chemotherapy/radiotherapy/immunotherapy and their corresponding dosages. The outcome can be thought of as the risk of cancer recurrence (Schwab et al., 2019).

**News:** The News dataset consists of word counts for news items. We extracted 10000 samples each with 2858 features. As in (Johansson et al., 2016; Schwab et al., 2019), we give meaning to our treatments and dosages by considering the treatment as being the viewing device (e.g. phone, tablet etc.) used to read the article and the dosage as being the amount of time spent reading it. The outcome can be thought of as user satisfaction.

**MIMIC III:** The Medical Information Mart for Intensive Care (MIMIC III) (Johnson et al., 2016) database consists of observational data from patients in the ICU. We extracted 3000 patients that receive antibiotics treatment and we used as features 9 clinical covariates measured during the day of ICU admission. Again, the features were scaled in the $[0, 1]$ interval. In this setting, we can considered as treatments the different antibiotics and their corresponding dosages.

For a summary description of the datasets, see table 6. The datasets are split into 64/16/20% for training, validation and testing respectively. The validation dataset is used for hyperparameter optimization.

|  | TCGA | News | MIMIC |
|---|---|---|---|
| Number of samples | 9659 | 10000 | 3000 |
| Number of features | 4000 | 2858 | 9 |
| Number of treatments | 3* | 3 | 2 |

Table 6: Summary description of datasets. *: for our final experiment in Appendix I.4 we increase the number of treatments in TCGA to 6 and 9.

## H  DOSAGE BIAS

In order to create dosage-assignment bias in our dataset, we assign dosages according to $d_w|\mathbf{x} \sim \text{Beta}(\alpha, \beta_w)$. The selection bias is controlled by the parameter $\alpha \geq 1$. When we set $\beta_w = \frac{\alpha-1}{d_w^*} + 2 - \alpha$ (which ensures that the mode of our distribution is $d_w^*$), we can write the variance of $d_w$ in terms of $\alpha$ and $d_w^*$ as follows

$$\text{Var}(d_w) = \frac{\frac{\alpha^2 - \alpha}{d_w^*} + 2\alpha - \alpha^2}{(\frac{\alpha-1}{d_w^*} + 2)^2(\frac{\alpha-1}{d_w^*} + 3)} \approx \frac{c\alpha^2}{d\alpha^3} . \tag{26}$$

We see that the variance of our Beta distribution therefore decreases with $\alpha$, resulting in the sampled dosages being closer to the optimal dosage, thus resulting in higher dosage-selection bias. In addition we note that the $\text{Beta}(1, 1)$ distribution is in fact the uniform distribution, corresponding to the dosages being sampled independently of the patient features, resulting in no selection bias when $\alpha = 1$.

# I ADDITIONAL RESULTS

## I.1 INVESTIGATING HYPERPARAMETER SENSITIVITY ($n_w$ AND $\lambda$)

Here we present additional results for our investigation of the hyperparameters $n_w$ and $\lambda$. Fig. 8 reports each of the 3 performance metrics as we increase the number of dosage samples, $n_w$, used to train the discriminators on the News dataset. As with the TCGA results in the main paper we see that the single discriminator suffers a significant performance decrease when $n_w$ is set too high.

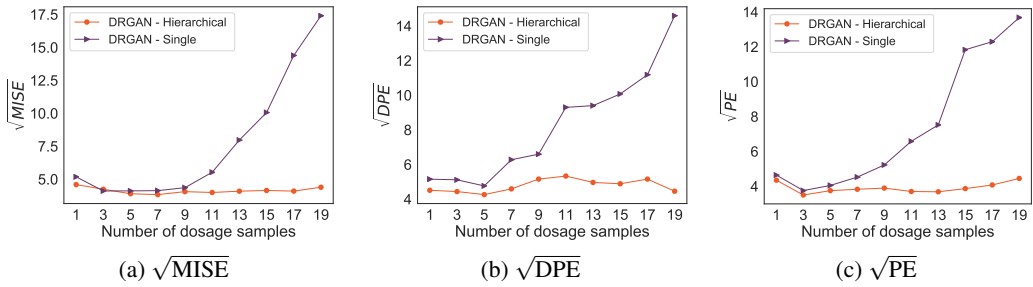

(a) $\sqrt{\text{MISE}}$       (b) $\sqrt{\text{DPE}}$       (c) $\sqrt{\text{PE}}$

Figure 8: Performance metrics when increasing the number of dosage samples on News dataset.

Fig. 9 and Fig. 10 report each of the performance metrics when we increase $\lambda$, the hyperparameter that trades off between the GAN loss, $\mathcal{L}$, and the supervised loss, $\mathcal{L}_S$. The results shown are on the TCGA and News datasets, respectively. As we would expect, for $\lambda = 0$, the performance is significantly worse than for $\lambda = 1$, since this corresponds to no supervised loss (this result is also backed up by our source of gain experiments). In addition, as we increase $\lambda$ we see performance slowly degrades, this is because the model starts to behave more and more like MLP-M since the supervised loss becomes dominant.

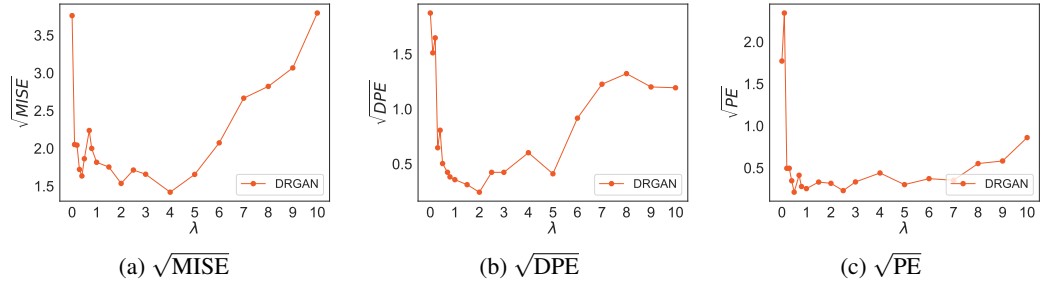

(a) $\sqrt{\text{MISE}}$       (b) $\sqrt{\text{DPE}}$       (c) $\sqrt{\text{PE}}$

Figure 9: Performance metrics when increasing $\lambda$ on TCGA dataset.

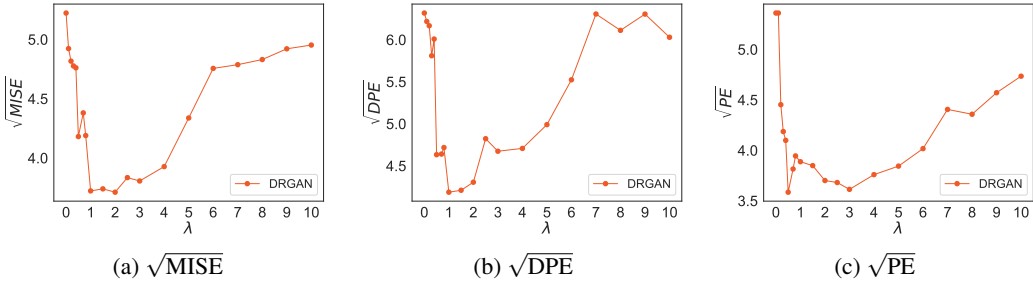

(a) $\sqrt{\text{MISE}}$       (b) $\sqrt{\text{DPE}}$       (c) $\sqrt{\text{PE}}$

Figure 10: Performance metrics when increasing $\lambda$ on News dataset.

## I.2 ADDITIONAL RESULTS ON SELECTION BIAS

In Fig. 11 we report the DPE and PE for our treatment and dosage bias experiment from Section 6.4 of the main paper.

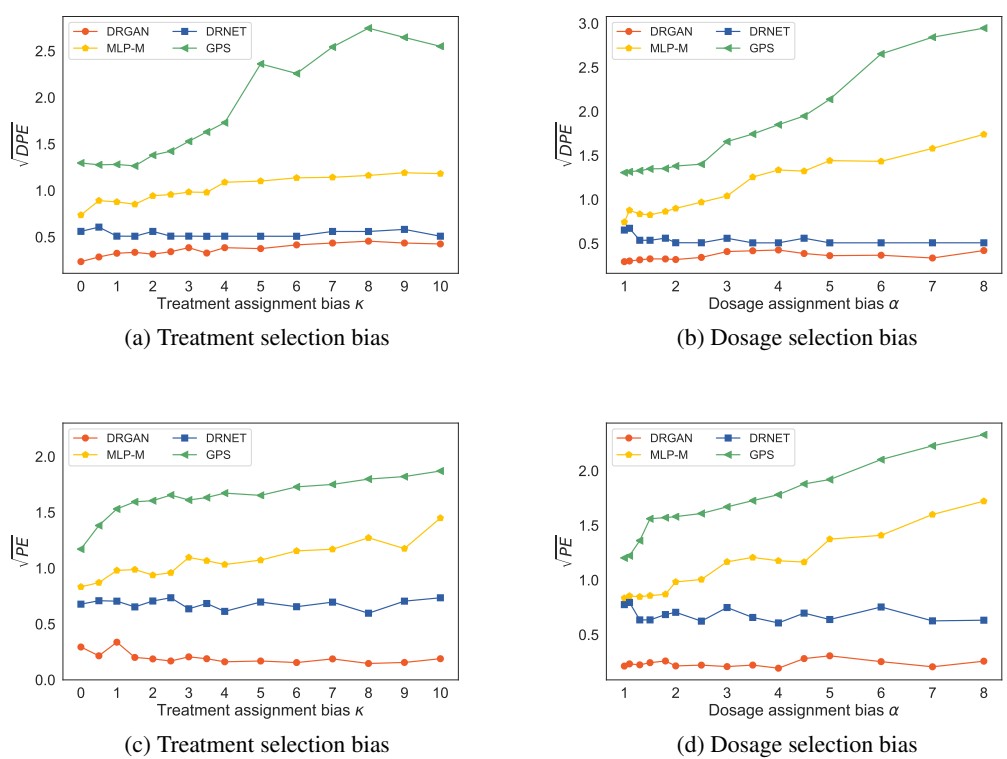

Figure 11: Additional performance metrics of the 4 methods on datasets with varying bias levels on TCGA dataset.

## I.3 DOSAGE POLICY ERROR FOR BENCHMARK COMPARISON

In Table 7 we report the Dosage Policy Error (DPE) corresponding to Section 6.3 in the main paper.

| Methods | TCGA $\sqrt{\text{DPE}}$ | News $\sqrt{\text{DPE}}$ | MIMIC $\sqrt{\text{DPE}}$ |
|---|---|---|---|
| **DRGAN** | $\mathbf{0.31} \pm 0.05$ | $\mathbf{4.14} \pm 0.11$ | $\mathbf{0.51} \pm 0.05$ |
| DRNet | $0.51 \pm 0.05^*$ | $4.39 \pm 0.11^*$ | $0.52 \pm 0.05$ |
| DRN-W | $0.50 \pm 0.05^*$ | $4.21 \pm 0.11$ | $0.53 \pm 0.05$ |
| GPS | $1.38 \pm 0.01^*$ | $6.40 \pm 0.01^*$ | $1.41 \pm 0.12^*$ |
| MLP-M | $0.92 \pm 0.05^*$ | $4.94 \pm 0.16^*$ | $0.77 \pm 0.05^*$ |
| MLP | $1.04 \pm 0.05^*$ | $5.18 \pm 0.12^*$ | $0.80 \pm 0.05^*$ |

Table 7: Performance of individualized treatment-dose response estimation on three datasets. Bold indicates the method with the best performance for each dataset. *: performance improvement is statistically significant.

### I.4 VARYING THE NUMBER OF TREATMENTS

In our final experiment, we increase the number of treatments by defining 3 or 6 additional treatments. The parameters $\mathbf{v}_1^w, \mathbf{v}_2^w, \mathbf{v}_3^w$ are defined in exactly the same way as for 3 treatments. The outcome shapes for treatments 4 and 7 are the same as for treatment 1, similarly for 5, 8 and 2 and for 6, 9 and 3. In Table 8 we report MISE, DPE and PE on the TCGA dataset with 6 treatments (TCGA-6) and with 9 treatments (TCGA-9). Note that we use 3 dosage samples for training DRGAN in this experiment.

| Method | TCGA - 6 | | | TCGA - 9 | | |
|---|---|---|---|---|---|---|
| | $\sqrt{\text{MISE}}$ | $\sqrt{\text{DPE}}$ | $\sqrt{\text{PE}}$ | $\sqrt{\text{MISE}}$ | $\sqrt{\text{DPE}}$ | $\sqrt{\text{PE}}$ |
| DRGAN | $2.37 \pm 0.12$ | $0.43 \pm 0.05$ | $0.32 \pm 0.05$ | $2.79 \pm 0.05$ | $0.51 \pm 0.05$ | $0.54 \pm 0.05$ |
| DRNET | $4.09 \pm 0.16$ | $0.52 \pm 0.05$ | $0.71 \pm 0.05$ | $4.31 \pm 0.12$ | $0.59 \pm 0.05$ | $0.74 \pm 0.05$ |
| GPS | $6.62 \pm 0.01$ | $2.04 \pm 0.01$ | $2.61 \pm 0.00$ | $7.58 \pm 0.01$ | $3.14 \pm 0.01$ | $2.91 \pm 0.01$ |

Table 8: Performance of DRGAN and the benchmarks when we increase the number of treatments in the dataset to 6 and 9. Bold indicates the method with the best performance for each dataset.

