# OpenReview forum: "Individualised Dose-Response Estimation using Generative Adversarial Nets"
_ICLR.cc/2020/Conference — Reject_

### Official Review · AnonReviewer3 · 2019-10-22
**Official Blind Review #3**

**Rating:** 1

**Review:**

The paper introduces Dose Response Generative Adversarial Network (DRGAN) that is aimed at generating entire dose-response curve from observational data with single dose treatments. This work is an extension of GANITE (Yoon et al., 2018) for the case of real-valued treatments (i.e., dosage). The proposed model consists of 3 blocks: (1) a generator, (2) a discriminator, and (3) an inference block. In this paper, GANITE’s generator and discriminator architectures are modified to be able to handle real-valued treatments.

This paper should be rejected due to the following arguments:
	- Since the paper is based on GANITE, I’m going to start my review by pointing out the problems/questions I have about GANITE. First, let me briefly summarize my understanding of this method:
		+ Given (x, t, yf), the generator G tries to estimate counterfactual outcomes (ycf).
		+ Given (x, (yf, ycf)), the discriminator D tries to figure out which outcome was factual.
		+ Yoon et al. (2018) claims that optimizing a GAN with this G and D, the generator will become better and better at estimating accurate counterfactual outcomes.
Looking at the objective function in Eq. (4), it is not clear why D should learn to distinguish factual from counterfactual outcome as opposed to learning the treatment selection bias as t.log(D(x ,y))+... would suggest. Yoon et al. (2018) assume the former while the latter seems more plausible. Of course, a by-product of learning the treatment selection bias is distinguishing yf from ycf, however, this doesn’t mean that G did a good job at generating an accurate estimation of ycf.
In summary, it seems that the adversarial training designed in GANITE and consequently DRGAN provide no advantage in terms of accurate estimation of counterfactuals, which in turn, nullifies the entire claimed contribution of these two works. I will, however, read the rebuttal carefully and am willing to modify the score if the authors address this major concern.

References:
	- Yoon, J., Jordon, J., & van der Schaar, M. (2018). GANITE: Estimation of individualized treatment effects using generative adversarial nets.


********UPDATE after reading the rebuttal********
Thank you for your response.
Still, there are multiple issues with this reasoning that you need to address.

$X$ is way more expressive than the added binary input that you described in the example. While the binary input is sampled from a Uniform distribution (either ½ or ⅔), $Pr(X|t=1)$ and $Pr(X|t=1)$ distributions are often Gaussian (certainly not Uniform) and therefore, embed a lot more information about which instances should receive which treatment. Therefore, $X$ is way more informative about selection bias than your binary input is about the image being fake/real. Do not underestimate this!!

It is true that the discriminator reaches its maximum loss when the generator’s estimation of the counterfactual is spot on. That is, it simply predicts the administered treatment according to the treatment selection bias. You need to show, however, that this maxima is global. Otherwise, there could be infinite maximas with the same loss; e.g., why not simply $Y^{factual} = Y^{counterfactual}$ does not reach this maxima?

You also need to show that the added information to $X$ by concatenating it with the factual outcome plus the generator’s estimation of the counterfactual outcome do improve prediction of the administered treatment -- i.e., minimizing the discriminator loss.

To summarize, I am not yet convinced that either GANITE or DRGAN do what they claim.



**Experience Assessment:**

I have published one or two papers in this area.

**Review Assessment: Checking Correctness Of Derivations And Theory:**

I carefully checked the derivations and theory.

**Review Assessment: Checking Correctness Of Experiments:**

I carefully checked the experiments.

**Review Assessment: Thoroughness In Paper Reading:**

I read the paper thoroughly.

---

> ### Author Response · Authors · 2019-11-08
> **Response for Reviewer 3**
>
> Thank you very much for your review!
>
> The objective function in Eq. (4) from GANITE will result in a discriminator that will use ALL discriminatory variables (within (x, y)) to distinguish factual from counterfactual outcomes. This means that while the discriminator will indeed use the treatment bias to distinguish between outcomes, it will also use the generated outcomes and how realistic they are to distinguish between them.
>
> When the generated outcomes are perfect (i.e. are being generated according to the true counterfactual distribution) then the discriminator will reduce to JUST using the treatment bias. However, if the generated outcomes are not good then the discriminator can do even better than the treatment bias by also incorporating the outcomes. Therefore, the generator can reduce its loss (with respect to the discriminator) by generating more realistic outcomes. Ultimately what is important is not what the discriminator can or cannot distinguish but rather what it can use to distinguish, and whether or not the generator can affect this by generating more realistic outcomes.
>
> To illustrate this point we can consider a toy example using vanilla GANs. Suppose that during the training of a vanilla GAN, in addition to giving as input to the discriminator the real or fake image, we also pass a binary input. For real images, we randomly generate this binary input by setting it to 0 or 1 with equal probability. For fake images, we do this by setting it to 0 with probability ⅔ and to 1 with prob. ⅓ . This setup directly emulates the treatment bias in our setting - this 0 or 1 cannot be affected by the generator (it depends only on whether the image is real or fake, not the pixel-values of the image itself, just as the treatment bias in GANITE and DRGAN depend only on the features and not on the generated counterfactuals). While the addition of these 0s and 1s might affect training (such as speed or stability), the optimal solution, which is for the generator to generate images from the true distribution is unchanged. As in our case, the min-max game played by the generator and discriminator will ultimately converge to a different value (rather than the -log4 found in the original GAN paper) but the learned generator will still be the same.
>
> We hope that our reply addresses your concern. Please let us know if further explanations are needed.

---

### Official Review · AnonReviewer2 · 2019-10-25
**Official Blind Review #2**

**Rating:** 3

**Review:**

The paper proposes a generative adversarial net model for heterogeneous dose-response causal effect estimation. The idea is to generate counterfactual dose-response curves using the generator that can fool the discriminator that tries to distinguish between factual data and counterfactual data. Factual data along with counterfactual data generated by the GAN can then be used for heterogeneous causal effect estimation.

To tailor to this specific heterogeneous dose-response curve estimation problem, the paper makes various architectural design modification from a regular GAN, such as using factual predictive performance as regularization, multitask for treatment option and dosage in the generator, hierarchical decomposition of the dose and response prediction in the discriminator, and permutation invariance when predicting the dosage-response curves for various treatment in the discriminator. These design choices are reasonably explained and the empirical gains due to these design choices are well documented (Table 2).

A few concerns:

1. It is unclear why using a GAN that seeks to produce samples that are indistinguishable from the factual ones will be the right way to synthesize counterfactual samples. Intuition and further justification need to be provided. Related to this concern, do the authors also implicitly reason under the consistency assumption (the observed/factual samples are the realizations potential outcomes of the corresponding treatment-dosage pairs in the potential outcome framework) ? If so, the authors should also reflect that in their list of assumptions.

2. It is also unclear what kind of theoretical guarantees the proposed method could deliver. Since the setting considered by the paper is to formally estimate the dosage-response effect, such guarantees, while potentially challenging to derive, are nonetheless crucial to better understand the circumstances under which the proposed method is reliable/unreliable.  This is especially the case given the elusive use of GAN to generate counterfactual outcomes as mentioned in the previous concern.

3. It will also be interesting to see whether the proposed method can be degenerated to handle simpler cases such as heterogeneous treatment effect estimation of binary treatment.  Or alternatively, when there is only one treatment option, where when treated there is a dosage-response curve while when not treated the dosage is zero. Related to this question, is this also the way to parameterize whether a subject receives a treatment or not in the model?



**Experience Assessment:**

I have read many papers in this area.

**Review Assessment: Checking Correctness Of Derivations And Theory:**

N/A

**Review Assessment: Checking Correctness Of Experiments:**

I assessed the sensibility of the experiments.

**Review Assessment: Thoroughness In Paper Reading:**

I read the paper at least twice and used my best judgement in assessing the paper.

---

> ### Author Response · Authors · 2019-11-08
> **Response for Reviewer 2**
>
> Thank you very much for your review!
>
> 1. We would first like to clarify that the generator is being given the task of generating outcomes that make identifying the factual one from among the factual+generated samples as hard as possible. This is conditional on the features and so it isn’t quite that the outcomes need to be indistinguishable (for example generating counterfactual outcomes that are identical to the factual outcome is not correct), but rather that the generated counterfactuals need to be as plausible as possible at being the factual outcome.
>
> This intuition is the same as in GANITE. Intuitively, if the generator is generating counterfactuals according to the true distribution of the counterfactuals then the “best” the discriminator can do is distinguish the factual treatment using the features alone (essentially guessing the correct treatment according to the treatment bias). When the generator is not generating counterfactuals according to the true distribution, then the discriminator will be able to use the generated outcomes as well as the features to distinguish among factual/counterfactual outcomes.
>
> We do reason under the consistency assumption and will update the manuscript to reflect this.
>
> 2. Theoretical guarantees for this work are particularly difficult to derive. Theoretical guarantees are not provided in the original GANITE paper. What can be shown for the original GANITE paper is that when the generator produces realistic outcomes, the discriminator reduces to predicting the treatment bias and this is a global minimum for the generator. This result is more difficult to prove in our setting, with dosages, but we believe it is also true and provides some intuition into why the GAN framework works for generating potential outcomes.
>
> 3. For the case of binary treatments our method reduces to the standard GANITE model. We are conducting an experiment in which there is only a single treatment and will report the results as soon as they are available. It is worth noting that in our experiments, a patient always receives some dosage of some treatment. The dosage parameter is normalised into [0,1] where 0 corresponds to the minimum dosage and 1 corresponds to the maximum dosage. However, the minimum dosage does not necessarily have to correspond to the treatment not being given, it may instead correspond to the smallest possible positive dosage that could be administered (perhaps due to practical reasons or due to clinical guidelines).
>
> If the possibility of no treatment needed to be considered, then this would be the same as incorporating an additional treatment that does not come with a dosage parameter - this can easily be done within our model. Such treatments will not need a dosage discriminator but will be passed to the treatment discriminator. A head can be added to the generator for each such non-dosage treatment but will not need to take dosage as an input.
>
> As for how treatments are encoded within our model, the factual treatment is one-hot encoded and is concatenated with the dosage parameter and other features. When generating each of the counterfactuals, the counterfactual to be generated is determined by choosing a head (for the treatment) and a dosage parameter to pass to the chosen head.
>
> We hope that our answers address your concerns. Please let us know if further explanations are needed.

---

> > ### Author Response · Authors · 2019-11-13
> > **Additional experimental results**
> >
> > In the following table, we report results for the binary treatment setup in which: treatment has an associated dosage (which requires estimating a dose-response curve) and no-treatment does not have an associated dosage (which requires estimating a single patient outcome).
> >
> > We generated data for the treatment outcome using $f_3(\mathbf{x}, d)$ (see Table 1 in the paper) and for the no-treatment outcome using $2C(\mathbf{v}_0^T \mathbf{x})$, where $\mathbf{v}_0$ are parameters, $\mathbf{x}$ are patient features and $C$ is the same scaling parameter used in the paper.
> >
> > As the DRNet public implementation does not allow for this set-up, we compared DRGAN with the multilayer perceptron model with multitask heads (MLP-M). This model is trained using supervised learning to minimize error on the factual outcomes and consists of two multitask heads: one head for the treatment option which receives as input the dosage and estimates the dose-response curve and one head for the no-treatment option.
> >
> > We report the MISE and the standard deviation for 10 runs of the models.
> >
> > ---------------------------------------------------------------------------------------------------
> > Dataset              |      TCGA - 2        |       News - 2          |     MIMIC - 2
> > ---------------------------------------------------------------------------------------------------
> >  DRGAN             |     1.27 ± 0.06      |      3.14 ± 0.13       |     0.65 ± 0.05
> >  MLP-M              |     2.01 ± 0.09      |      4.65 ± 0.08       |     1.52 ± 0.06
> > ---------------------------------------------------------------------------------------------------
> >
> > As can be seen in the table, DRGAN is capable of handling this setting and lends itself naturally to potentially mixed dosage and no-dosage treatment options.

---

### Official Review · AnonReviewer1 · 2019-10-29
**Official Blind Review #1**

**Rating:** 3

**Review:**

Summary:
Treatment Response Estimation
Different from Average treatment effect because of multiple treatment and dosage parameters the problem gets complicated
Previously, other papers had tried to estimate ITE using distribution matching, propensity dropout  and matching on balancing scores.
Most relevant benchmark DrNet(https://arxiv.org/pdf/1902.00981.pdf)
Here, authors use GANs to perform the task of Dose-response estimation instead of traditional methods. They use a counterfactual generator and counterfactual discriminator for the task. Generator is modeled as a multitask neural network and the discriminator is modeled as hierarchical network consisting of treatment and dosage discriminator.
Overall, the model outperforms other baselines such as DRNet and GPS indicating improved dose-response estimation capabilities.
Pros:
Significant empirical advance? Did they solve a standing open problem?
Significant performance improvement   in defined metrics compared to previous methods such DrNet. This indicates the proposed methods are better for dose-response estimation.
Any good practical outcome (code, algorithm, etc)?
Novel neural network architectures used with counterfactual generator and counterfactual discriminator are interesting and these can be extended to other tasks.
Are the experiments well executed?
Good experiments in general. Source of gain experiment(section 6.2) provides deeper insight into why the model might perform better compared to other baselines.

Useful for the community in general?
Yes it’s useful for the community as this is a challenging problem and estimating dose-response from observational data(EHR) is tricky.

Cons:
Any missing baselines? Missing datasets?
Missing baselines: TARNET, MLP, GANITE, kNN, BART. These baselines were evaluated in DrNet paper.
Any odd design choices in the algorithm not explained well?
Why did authors use GAN with Vanilla loss function while it has been shown that there are other loss functions which can help in improving the performance? Perhaps evaluating the GAN with other loss function can provide further insight to the stability of the model

Quality of writing?
The overall flow of the paper is a little hard to follow.

Is the model better in terms of stability?
Compared to DrNet Paper, is the model easy to train? Since the model uses a GAN loss function which are known for being hard to train, it would be good to evaluate the model in terms of difficulty of training.

Is there sufficient novelty in what they propose? Minor variation of previous work?
While this is a very interesting task, this paper falls under incremental improvement as it improves GANITE framework to solve dose-response estimation. As authors use GAN for counterfactual estimation which is not novel(GANITE) and estimating dose-response estimation is not novel(DrNet).
Specific Points
In comparison to DrNet Paper(Learning Counterfactual Representations for Estimating Individual Dose-Response Curves)
Square root of MISE is different for the datasets.
Is it because of different hyperparameters?

**Experience Assessment:**

I have read many papers in this area.

**Review Assessment: Checking Correctness Of Derivations And Theory:**

I did not assess the derivations or theory.

**Review Assessment: Checking Correctness Of Experiments:**

I assessed the sensibility of the experiments.

**Review Assessment: Thoroughness In Paper Reading:**

I made a quick assessment of this paper.

---

> ### Author Response · Authors · 2019-11-08
> **Initial response for Reviewer 1**
>
> Thank you very much for your review!
>
> * Novelty *
>
> We would like to address your comment about the contributions and improvements this paper brings.
>
> To begin with, individualized dose-response estimation represents a significantly more complex problem than estimating individualized effects of binary and categorical treatments. Compared to the binary treatment setting where only one counterfactual needs to be estimated (see Figure 1), for the dose-response setting, a potentially infinite number of counterfactuals need to be estimated (due to the continuous nature of the treatment dosage).
>
> GANITE can only be used to estimate counterfactuals for multiple treatments. Our method (DRGAN) uses the adversarial training idea from GANITE as a building block. However, it proposes significant technical improvements to be able to estimate individualized dose-response curves.
>
> One important difference is that the GANITE discriminator receives as input all generated potential outcomes. However, for DRGAN, it would not be sensible to give to the discriminator as input entire dose-response curves (as discussed on page 2 in the paper). Instead, for the DRGAN discriminator, we propose sampling points from the dose-response curves and ask the discriminator to pick the factual outcome from these samples. This reduces complexity while still ensuring that the dose-response curve is well estimated.
>
> In order for the discriminator to handle the complexity of identifying the factual dosage from the samples given as input, we also propose the following design choices:
>
> 1. The discriminator uses a hierarchical architecture consisting of a treatment discriminator and dosage discriminators.
>
> 2. We use permutation invariant/equivariant layers in the treatment and dosage discriminators to directly model them as functions on sets.
>
> Moreover, while the problem of individualized dose-response estimation in itself is not novel that does not mean that methods should not be developed to solve it as well as possible. In fact, our experimental results show significant improvements over DRNet and other methods.
>
> We will improve the explanations in the paper to highlight these contributions better.
>
>
> * Vanilla GAN loss *
>
> The vanilla GAN loss (i.e. the one that results in learning via the Jensen-Shannon divergence) was used in the original GANITE paper and demonstrated good performance. Such a loss has natural intuition in the case of multiple treatments (or multiple dosages) when the objective becomes to select from among many different outcomes the single correct one. Defining other well-studied GAN losses (such as WGAN) is not trivial as what is essentially happening in our setting is not that we are comparing two distributions (real outcomes vs fake outcomes) but multiple distributions (real outcome vs each generated outcome). We are also not comparing entire samples but rather identifying a single outcome from among a mixed real/fake sample.
>
>
> * Model stability *
>
> In our experiments, we found both the DRGAN and DRNET models stable during train. However, we believe that the main metric that should be used for evaluating the models is the error in estimating individualized dose-response curves. Once the models are trained and deployed in real-world scenarios it is more important that they perform well rather than that they were easy to train in the first place.

---

> > ### Author Response · Authors · 2019-11-08
> > **Additional response for Reviewer 1**
> >
> > * Baselines *
> >
> > We use a Multilayer Perceptron (MLP) as a baseline for comparison (see Table 3). Moreover, our Multilayer Perceptron with multitask heads (MLP-M) follows a similar model structure with TARNET: it consists of several neural network layers shared between all treatments followed by a multitask head for each treatment. The dosage is given as input to each multitask head. We will mention in the paper that this represents an extension of TARNET to handle dosages. The results for this baseline are also in Table 3.
> >
> > However, note that GANITE, kNN, BART are causal inference methods for categorical treatments and cannot be extended to handle dosages in a sensible way. For example, Ho et al. [1] proposed kNN and in Section 2.4 they mention that in the case of continuous treatments (such as dosage) other methods designed for this task should be used instead since kNN is designed for binary treatments.
> >
> > In fact, the DRNET paper (page 7) also mentions that these models were naively extended by including the dosage as a patient covariate. Please also note that DRNET has outperformed this naive extension of these baselines in the dose-response setting.
> >
> >
> > * Experimental results *
> >
> > While the features used for the News and TCGA datasets are similar to the ones in the DRNet paper, we used different semi-synthetic data generation model which is why the specific numerical results between the papers are different.
> >
> > Since individualized dose-response estimation has been a less-studied problem in the causal inference literature, there are no standard benchmark datasets for comparing models. Our proposed semi-synthetic data generation model also represents a contribution of the paper since it introduces such a benchmark. The proposed method for semi-synthetic data generation allows for modelling both treatment and dosage selection bias and it incorporates different shapes for the dose-response curves.
> >
> >
> > We hope that our reply has addressed your concerns! Please let us know if further clarifications are needed.
> >
> > [1] Daniel E Ho, Kosuke Imai, Gary King, and Elizabeth A Stuart. Matching as nonparametric preprocessing for reducing model dependence in parametric causal inference

---

### Decision · Program_Chairs · 2019-12-19

**Decision:**

Reject

**Comment:**

This paper addresses the problem of estimating treatment responses involving a continuous dosage parameter.  The basic idea is to learn a GAN model capable of generating synthetic dose-response curves for each training sample, which then facilitates the supervised training of an inference model that estimates these curves for new cases.  For this purpose, specialized architectures are also proposed for the GAN, which involves a multi-task generator network and a hierarchical discriminator network.  Empirical results demonstrate improvement over existing methods.

While there is always a chance that reviewers may underappreciate certain aspects of a submission, the fact that there was a unanimous decision to reject this work indicates that the contribution must be better marketed to the ML community.  For example, after the rebuttal one reviewer remained unconvinced regarding explanations for why the proposed method is likely to learn the full potential outcome distribution.  Among other things, another reviewer felt that both the proposed DRGAN model, and the GANITE framework upon which it is based, were not necessarily working as advertised in the present context.